# Understanding the Effectiveness of Lipschitz-Continuity in Generative Adversarial Nets

## Abstract

In this paper, we investigate the underlying factor that leads to the failure and success in training of GANs. Specifically, we study the property of the optimal discriminative function $f^*(x)$ and show that $f^*(x)$ in most GANs can only reflect the local densities at $x$, which means the value of $f^*(x)$ for points in the fake distribution ($P_g$) does not contain any information useful about the location of other points in the real distribution ($P_r$). Given that the supports of the real and fake distributions are usually disjoint, we argue that such a $f^*(x)$ and its gradient tell nothing about "how to pull $P_g$ to $P_r$", which turns out to be the fundamental cause of failure in training of GANs. We further demonstrate that a well-defined distance metric (including the dual form of Wasserstein distance with a compacted constraint) does not necessarily ensure the convergence of GANs. Finally, we propose Lipschitz-continuity condition as a general solution and show that in a large family of GAN objectives, Lipschitz condition is capable of connecting $P_g$ and $P_r$ through $f^*(x)$ such that the gradient $\nabla_x f^*(x)$ at each sample $x \sim P_g$ points towards some real sample $y \sim P_r$.

## 1 Introduction

Generative Adversarial Networks (GANs) (Goodfellow et al., 2014), as a new way of learning generative models, have recently shown promising results in various challenging tasks. Although GANs are popular and widely-used (Isola et al., 2016; Brock et al., 2016; Nguyen et al., 2016; Zhu et al., 2017; Karras et al., 2017), they are notoriously hard to train (Goodfellow, 2016). The underlying obstacles, though have been widely studied (Arjovsky & Bottou, 2017; Lucic et al., 2017; Heusel et al., 2017a; Mescheder et al., 2017; 2018; Yadav et al., 2017), are still not fully understood. In this paper, we study the convergence of GANs from the perspective of the optimal discriminative function $f^*(x)$.

We show that in original GAN and its most variants, $f^*(x)$ is a function of densities at the current point $x$ but does not reflect any information about the densities/locations of other points in the real and fake distributions. Moreover, Arjovsky & Bottou (2017) state that the supports of real and fake distributions are usually disjoint. In this paper, we argue that the fundamental cause of failure in training of GANs (Section 2.1) stems from the combination of the above two facts. The generator uses $\nabla_x f^*(x)$ as the guidance for updating the generated samples, but $\nabla_x f^*(x)$ actually tells nothing about where $P_r$ is. Therefore, the generator is not guaranteed to converge to the case $P_g = P_r$.

Accordingly, Arjovsky et al. (2017) proposed the Wasserstein distance (in its dual form) as an alternative objective, which can properly measure the distance between two distributions no matter whether their supports are disjoint. However, as shown in Section 2.3, when the supports of the $P_g$ and $P_r$ are disjoint, the gradient of $f^*(x)$ from the dual form of Wasserstein distance given a compacted dual constraint also does not reflect any useful information about other points in $P_r$. Based on this observation, we provide further investigation in Section 2.4 and argue that measuring the distance properly does not necessarily imply that the gradient is well-defined in terms of $\nabla_x f^*(x)$.

In Section 3, we propose incorporating Lipschitz-continuity condition in the objectives of GANs as a general solution, and prove that in a **broad** family of discriminator objectives, Lipschitz-continuity condition can build strong connections between $P_g$ and $P_r$ through $f^*(x)$ such that when the supports of $P_g$ and $P_r$ are disjoint, $\nabla_x f^*(x)$ at each sample $x \sim P_g$ will point towards some real sample $y \sim P_r$. This guarantees that $P_g$ is moving towards $P_r$ at every step.

We extend our discussion on $f^*(x)$ and $\nabla_x f^*(x)$ to the case where the supports of $P_g$ and $P_r$ are overlapped, and present the general arguments in Section 4.1. Subsequently, in Section 4.2, we show that the locality of $f^*(x)$ and $\nabla_x f^*(x)$ in traditional GANs turns out to be an intrinsic cause to mode collapse. Finally, in Section 4.3, we explain the reason of empirical success of traditional GANs under the circumstance that they have no convergence guarantee.

Table 1: Comparison of different objectives in GAN models.

| | $\phi$ | $\varphi$ | $\mathcal{F}$ | $f^*(x)$ |
|---|---|---|---|---|
| JS-Divergence | $-\log(\sigma(-x))$ | $-\log(\sigma(x))$ | $\{f : \mathbb{R}^n \to \mathbb{R}\}$ | $\log \frac{P_r(x)}{P_g(x)}$ |
| Least Square | $(x-\alpha)^2$ | $(x-\beta)^2$ | $\{f : \mathbb{R}^n \to \mathbb{R}\}$ | $\frac{\alpha \cdot P_g(x) + \beta \cdot P_r(x)}{P_g(x) + P_r(x)}$ |
| Wasserstein-1 with Lip$_1$ | $x$ | $-x$ | $\{f : \mathbb{R}^n \to \mathbb{R}, \|f\|_{lip} \le 1\}$ | $N/A$ |
| $\mu$-Fisher IPM | $x$ | $-x$ | $\{f : \mathbb{R}^n \to \mathbb{R}, \mathbb{E}_{x\sim\mu}\|f(x)\|^2 \le 1\}$ | $\frac{1}{\mathcal{F}_\mu(P_r, P_g)} \frac{P_r(x) - P_g(x)}{\mu(x)}$ |

## 2 THE FUNDAMENTAL CAUSE OF FAILURE IN TRAINING OF GANS

Typically, the objectives of GANs can be formulated as follows:

$$\min_{f \in \mathcal{F}} J_D \triangleq \mathbb{E}_{z \sim P_z}[\phi(f(g(z)))] + \mathbb{E}_{x \sim P_r}[\varphi(f(x))],$$
$$\min_{g \in \mathcal{G}} J_G \triangleq \mathbb{E}_{z \sim P_z}[\psi(f(g(z)))], \tag{1}$$

where $P_z$ is the source distribution of the generator (usually a Gaussian distribution) in $\mathbb{R}^m$ and $P_r$ is the target (real) distribution in $\mathbb{R}^n$. The generative function $g : \mathbb{R}^m \to \mathbb{R}^n$ learns to output samples that shares the same dimension as $P_r$, while the discriminative function $f : \mathbb{R}^n \to \mathbb{R}$ learns to output a score indicating the authenticity of a given sample. We denote the implicit distribution of the generated samples as $P_g$, i.e., $P_g = g(P_z)$.

$\mathcal{F}$ and $\mathcal{G}$ denote discriminative and generative function spaces parameterized by neural networks, respectively; functions $\phi, \varphi, \psi : \mathbb{R} \to \mathbb{R}$ are loss metrics. We list the choices of $\mathcal{F}, \phi$ and $\varphi$ in some representative GAN models in Table 1, where we denote $f^* = \arg\min_{f \in \mathcal{F}} J_D$.

In these GANs, the gradient that the generator receives from the discriminator with respect to a generated sample $x \sim P_g$ is

$$\nabla_x J_G(x) = \nabla_{f(x)} \psi(f(x)) \cdot \nabla_x f(x). \tag{2}$$

In Eq. (2), the first term $\nabla_{f(x)} \psi(f(x))$ is a step-related scalar that is out of the scope of our discussion in this paper; the second term $\nabla_x f(x)$ is a vector indicating the direction that the generator should follow for optimizing on sample $x$.

### 2.1 $\nabla_x f^*(x)$ ON $P_g$ DOES NOT REFLECT USEFUL INFORMATION ABOUT $P_r$

In this section, we will show that when the supports of $P_g$ and $P_r$ are disjoint, $\nabla_x f^*(x)$ in traditional GANs does not reflect any useful information about $P_r$, and $P_g$ is not guaranteed to converge to $P_r$. We argue that this is the fundamental cause of non-convergence and instability in traditional GANs.[1]

#### 2.1.1 THE ORIGINAL GAN AND LEAST-SQUARES GAN

In the simplest case of Eq. (1), e.g., the original GAN (Goodfellow et al., 2014) and Least-Squares GAN (Mao et al., 2016), there is no restriction on $\mathcal{F}$. Therefore, $f^*(x)$ for each point $x$ is independent of other points, and we have

$$f^*(x) = \arg\min_{f(x) \in \mathbb{R}} P_g(x) \cdot \phi(f(x)) + P_r(x) \cdot \varphi(f(x)), \forall x. \tag{3}$$

Since we assume supports of $P_g$ and $P_r$ are disjoint, we further have

$$f^*(x) = \begin{cases} \arg\min_{f(x) \in \mathbb{R}} P_g(x) \cdot \phi(f(x)), \forall x \sim P_g, \\ \arg\min_{f(x) \in \mathbb{R}} P_r(x) \cdot \varphi(f(x)), \forall x \sim P_r. \end{cases} \tag{4}$$

---

[1]In this paper, traditional GANs mainly refers to the original GAN and Least-Squares GAN, where $f^*(x)$ depends only on the densities $P_g(x)$ and $P_r(x)$. Broadly, it refers to all GANs where $f^*(x)$ does not reflect information about the locations of the other points in $P_g$ and $P_r$, such as the Fisher GAN.

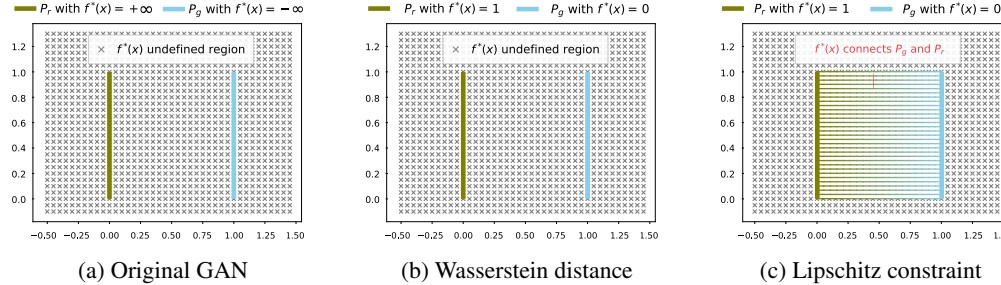

(a) Original GAN  (b) Wasserstein distance  (c) Lipschitz constraint

Figure 1: In traditional GANs, $f^*(x)$ is only defined on the supports of $P_g$ and $P_r$ and its values do not reflect any information about the locations of other points in $P_g$ and $P_r$. Therefore, they have no guarantee on the convergence. Wasserstein distance in a compacted dual form suffers from the same problem. GANs under Lipschitz constraint builds connection between $P_g$ and $P_r$, where $\nabla_x f^*(x)$ pulls $P_g$ towards $P_r$.

For $x \sim P_g$, the value of $f^*(x)$ is irrelevant to $P_r$. Since $P_g$ and $P_r$ are disjoint[2], $\nabla_x f^*(x)$ for $x \sim P_g$ also tells nothing about $P_r$. In consequence, the generator can hardly learn useful information and is not guaranteed to converge to the case where $P_g = P_r$.

### 2.1.2 THE FISHER GAN

Mroueh et al. (2017) prove that the optimal $f^*$ of $\mu$-Fisher IPM $\mathcal{F}_\mu(P_r, P_g)$, the objective used in Fisher GAN (Mroueh & Sercu, 2017), has the following form

$$f^*(x) = \frac{1}{\mathcal{F}_\mu(P_r, P_g)} \frac{P_r(x) - P_g(x)}{\mu(x)}. \tag{5}$$

where $\mu$ is a distribution whose support covers $P_r$ and $P_g$. Given $P_r$ and $P_g$ are disjoint, we have

$$f^*(x) = \begin{cases} \frac{1}{\mathcal{F}_\mu(P_r, P_g)} \frac{-P_g(x)}{\mu(x)}, & \forall x \sim P_g; \\ \frac{1}{\mathcal{F}_\mu(P_r, P_g)} \frac{P_r(x)}{\mu(x)}, & \forall x \sim P_r; \\ 0, & \text{otherwise.} \end{cases} \tag{6}$$

Note that the scalar $\frac{1}{\mathcal{F}_\mu(P_r, P_g)}$ is a constant. Eq. (6) also defines $f^*(x)$ on $P_g$ and $P_r$ independently. Therefore, for $x \sim P_g$, $f^*(x)$ and $\nabla_x f^*(x)$ tell nothing about $P_r$.

### 2.2 CONNECTION TO GRADIENT VANISHING

The non-convergence problem of the original GAN has once been considered as the gradient vanishing problem. In (Goodfellow et al., 2014), it is addressed by using an alternative objective for the generator. However, it actually only changes the scalar $\nabla_{f(x)} \psi(f(x))$ while the aforementioned problem in $\nabla_x f^*(x)$ still exists. The least-squares GAN (Mao et al., 2016) is proposed to address the gradient vanishing problem, but it also focuses on $\nabla_{f(x)} \psi(f(x))$ basically. As we have discussed, the least-squares GAN also belongs to traditional GANs, which is not guaranteed to converge when $P_g$ and $P_r$ are disjoint.

Arjovsky et al. (2017) provided a new perspective on understanding the gradient vanishing problem. They argued that gradient vanishing stems from the ill-behaving of traditional metrics, i.e., the distance between $P_g$ and $P_r$ remains constant when they are disjoint. Wasserstein distance is thus proposed as an alternative metric, which can properly measure the distance between two distributions no matter they are disjoint or not. However, as we will show next, Wasserstein distance may also suffer from the same problem on $\nabla_x f^*(x)$, if a more compact dual form is used.

In summary, gradient vanishing is about the scalar term $\nabla_{f(x)} \psi(f(x))$ in $\nabla_x J_G(x)$ or the overall scale of $\nabla_x J_G(x)$, and in this paper we investigate its direction $\nabla_x f^*(x)$, where the problem is so fundamental and challenging that even the Wasserstein distance, which can properly measure the distance for disjoint distributions, may also suffer from the same issue.

---

[2]Here and later, "two distributions are disjoint" means that their supports are disjoint.

## 2.3 WASSERSTEIN DISTANCE IN COMPACT DUAL FORM SUFFERS FROM THE SAME PROBLEM

The ($1^{st}$-)Wasserstein distance is a distance function defined between two probability distributions:

$$W_1(P_r, P_g) = \inf_{\pi \in \Pi(P_r, P_g)} \mathbb{E}_{(x,y) \sim \pi} [d(x, y)], \tag{7}$$

where $\Pi(P_r, P_g)$ denotes the collection of all probability measures with marginals $P_r$ and $P_g$ on the first and second factors, respectively. Since solving it in the primal form (Eq. (7)) is burdensome, Wasserstein distance is usually solved in its dual form. Though Wasserstein distance in its dual form is usually written with Lipschitz constraint, we here provide a more compact version. The proof of this dual form can be found in Appendix I.

$$W_1(P_r, P_g) = \sup_f \mathbb{E}_{x \sim P_r} [f(x)] - \mathbb{E}_{x \sim P_g} [f(x)],$$
$$s.t. \ f(x) - f(y) \leq d(x, y), \ \forall x \sim P_r, \forall y \sim P_g. \tag{8}$$

We leave the detailed discussion on the relationship between Lipschitz-continuity and Wasserstein distance in Section 4.4. In Eq. (8), we replace the strong Lipschitz constraint with a looser constraint. Note that Eq. (7) and Eq. (8) are still equivalent. With this dual form, we will demonstrate that a well-defined distance metric, i.e., Wasserstein distance in this compacted dual form, may also suffer from the same problem in $\nabla_x f^*(x)$ and does not necessarily ensure the convergence of GANs.

We now study the optimal discriminative function $f^*(x)$ of Wasserstein distance in this dual form. Since there is generally no closed-form solution for $f^*(x)$ in Eq. (8), we use an illustrative example for demonstration here, but the conclusion is general. Let $Z \sim U[0, 1]$ be a uniform variable on interval $[0, 1]$, $P_g$ be the distribution of $(1, Z) \in \mathbb{R}^2$, and $P_r$ be the distribution of $(0, Z) \in \mathbb{R}^2$, as shown in Figure 1. According to Eq. (8), one of the optimal $f^*$ is as follows

$$f^*(x) = \begin{cases} 0 & \forall x \sim P_g, \\ 1 & \forall x \sim P_r. \end{cases} \tag{9}$$

Though having the constraint "$f(x) - f(y) \leq d(x, y), \forall x \sim P_r, \forall y \sim P_g$", Wasserstein distance in this dual form also only defines the value of $f^*(x)$ on the supports of $P_g$ and $P_r$, and the values of $f^*(x)$ on $P_g$ contain no useful information about the location of $P_r$. Therefore, if $P_g$ and $P_r$ are disjoint, $\nabla_x f^*(x)$ hardly provides useful information to the generator about "how to change $P_g$ into $P_r$" and the generator is not guaranteed to converge to the case $P_g = P_r$. It is worth noticing that the value of $f^*(x)$ on the region other than the supports of $P_g$ and $P_d$ is still undefined.

## 2.4 A WELL-DEFINED METRIC DOES NOT NECESSARILY GUARANTEE THE CONVERGENCE

The objectives of GANs are usually defined as (or proved equivalent to) minimizing a distance metric between $P_g$ and $P_r$, which implies that "$P_g = P_r$ is the unique global optimum", and is in accordance with the final goal of the generative model, i.e., estimating the distribution of real samples. However, in this section, we emphasize that a smooth distance metric satisfying "$P_g = P_r$ is the optimum" does not necessarily guarantee the convergence of GANs.

Given an objective is convex with respect to $P_g$ and holds the property that "$P_g = P_r$ is the unique optimum", the convergence of GANs is guaranteed if only they are directly optimizing $P_g$, i.e., "dragging" $P_g$ to $P_r$. However, what they actually do is using $\nabla_x f^*(x)$ as the guidance to optimize the generated samples. As shown in previous sections, when $P_g$ and $P_r$ are disjoint, $\nabla_x f^*(x)$, i.e., the direction that the generator follows for updating the generated samples, tells nothing about how to pull $P_g$ to $P_r$. In this way, the convergence of GANs are not necessarily guaranteed. $\nabla_x f^*(x)$ indeed indicates the direction of decreasing the objective in terms of the current $f^*(x)$, but updating $x$ to make the value of $f^*(x)$ increase / decrease does not necessarily imply that $P_g$ is getting closer to $P_r$. Recall that in the failure case of Wasserstein distance dual form in the above section, the values of $f^*(x)$ on $P_g$ is $zero$, while the values of $f^*(x)$ around $P_g$ is undefined.

In conclusion, a smooth distance metric satisfying "$P_g = P_r$ is the optimum" does not guarantee the convergence and sample updating according to $\nabla_x f^*(x)$ does not necessarily decrease the distance between $P_g$ and $P_r$. Therefore, if we use $\nabla_x f^*(x)$ for updating the generator, it is necessary to make $\nabla_x f^*(x)$ aware of how to pull $P_g$ to $P_r$. Alternative strategies actually exist, for example, Sanjabi et al. (2018) use the optimal transport plan (Seguy et al., 2017) between $P_g$ and $P_r$ to update the generator. However, their results tend to be blurry. In the next section, we will introduce the Lipschitz constraint as a general solution for making $\nabla_x f^*(x)$ well-behaving and guaranteeing the convergence of $\nabla_x f^*(x)$-based GANs.

## 3  A GENERAL SOLUTION: LIPSCHITZ CONDITION

Lipschitz-continuity constraint becomes popular in GANs recently as part of the discriminator's objective (Arjovsky et al., 2017; Kodali et al., 2017; Fedus et al., 2017; Miyato et al., 2018), achieving great success. In this section, we explain the significance of Lipschitz constraint when introduced into the objective of the discriminator. In a nutshell, under a board family of GAN objectives, Lipschitz constraint is able to connect $P_g$ and $P_r$ through $f^*(x)$ such that when $P_r$ and $P_g$ are disjoint, $\nabla_x f^*(x)$ for each generated sample $x \sim P_g$ will point towards some real sample $y \sim P_r$, which guarantees the trend that "$P_g$ is getting closer to $P_r$ at every step".

### 3.1  THE MAIN RESULT

A function $f : X \to Y$ is $k$-Lipschitz continuous if it satisfies the following property:

$$d_Y(f(x), f(y)) \leq k \cdot d_X(x, y), \forall\, x, y \in X, \tag{10}$$

where $d_X$ and $d_Y$ are distances metrics in domains $X$ and $Y$, respectively. The smallest constant $k$ is called the Lipschitz constant of function $f$. In this paper (and most GAN papers), $d_X$ and $d_Y$ are defined as Euclidean distance.[3] We let $\|y-x\|$ denote Euclidean distance.

As proved by Gulrajani et al. (2017), when the Lipschitz condition is combined with Wasserstein distance, we have the following property if $f^*(x)$ is differentiable, then

$$\Pr\left(\nabla_x f^*(x_t) = \frac{y - x}{\|y - x\|}\right) = 1, \text{ for } (x, y) \sim \pi^*, \tag{11}$$

where $x_t = tx + (1 - t)y$, $0 \leq t \leq 1$, and $\pi^*$ is the optimal $\pi$ in Eq. (7). The meaning of this proposition is two-fold: (i) for each $x \sim P_g$, there exists a $y \sim P_r$ such that $\nabla_x f^*(x_t) = \frac{y-x}{\|y-x\|}$ for all linear interpolations $x_t$ between $x$ and $y$; (ii) these $(x, y)$ pairs match the optimal coupling $\pi^*$.

Next we introduce our theorem on the Lipschitz condition. It turns out when combining the Lipschitz condition with generalized objectives, Property-(i) still holds and Property-(ii) is naturally dismissed as it is now not restricted to Wasserstein distance.

**Theorem 1.** Let $J_D \triangleq \mathbb{E}_{x \sim P_g}[\phi(f(x))] + \mathbb{E}_{x \sim P_r}[\varphi(f(x))]$ and $\partial_x J_D$ denotes $P_g(x)\phi(f(x)) + P_r(x)\varphi(f(x))$. Let $\bar{P}_r$ and $\bar{P}_g$ denote the supports of $P_r$ and $P_g$, respectively. Assume $f^* = \arg\min_f [J_D + \lambda \cdot k(f)^2]$, where $k(f)$ is the Lipschitz constant of $f$. If $\phi(x)$ and $\varphi(x)$ in $J_D$ satisfy

$$\begin{cases} \phi'(x) > 0, \phi''(x) \geq 0, \\ \varphi'(x) < 0, \varphi''(x) \geq 0, \\ \exists\, a,\ \phi'(a) + \varphi'(a) = 0, \end{cases} \tag{12}$$

then we have that

(a)  $\forall x \in \bar{P}_g \cup \bar{P}_r, \exists y_{\neq x} \in \bar{P}_g \cup \bar{P}_r$ such that $|f^*(y) - f^*(x)| = k(f^*) \cdot \|x - y\|$ or $\nabla_{f^*(x)} \partial_x J_D = 0$;

(b)  $\forall x \in \bar{P}_g \cup \bar{P}_r - \bar{P}_g \cap \bar{P}_r, \exists y_{\neq x} \in \bar{P}_g \cup \bar{P}_r$ such that $|f^*(y) - f^*(x)| = k(f^*) \cdot \|x - y\|$;

(c)  if $\bar{P}_g = \bar{P}_r$ and $P_g \neq P_r$, then $\exists x, \exists y_{\neq x}$ such that $|f^*(y) - f^*(x)| = k(f^*) \cdot \|x - y\|$;

(d)  the only Nash Equilibrium of $J_D + \lambda \cdot k(f)^2$ is reached when $P_g = P_r$, where $k(f) = 0$.

The above theorem states that when the Lipschitz condition is combined with an objective that satisfies Eq. (12), then: (a) for the optimal discriminative function $f^*(x)$ at any point $x \in \bar{P}_g \cup \bar{P}_r$, it either is bounded by the Lipschitz constant or $\partial_x J_D$ holds a zero-gradient with respect to $f^*(x)$; (b) for any point that only appears in $\bar{P}_g$ or $\bar{P}_r$, there must exist a point that bounds this point in terms of $|f^*(y) - f^*(x)| = k(f^*) \cdot \|x - y\|$, because for these points, $\partial_x J_D$ will never get zero gradient with respect to $f^*(x)$ as we prove in the Appendix G; (c) when $P_g$ and $P_r$ are totally overlapped, as long as $P_g$ still not converges to $P_r$, there exists at least one pair $(x, y)$ that bounds each other; (d) the only Nash Equilibrium among $P_g$ and $f^*(x)$ under this objective is "$P_g = P_r$ with $k(f^*) = 0$". The formal proof is in Appendix G.

---

[3]Actually, we argue that the distance metrics must be Euclidean distance in GANs. See Appendix D.

Wasserstein distance, i.e., $\phi(x) = \varphi(-x) = x$ is one instance that satisfies Eq. (12); and it is a very special case, which holds $\phi''(x) = 0$ and $\varphi''(x) = 0$. Eq. (12) is actually quite general and there exists many other settings, e.g., $\phi(x) = \varphi(-x) = -\log(\sigma(-x))$, $\phi(x) = \varphi(-x) = x + \sqrt{x^2 + 1}$ and $\phi(x) = \varphi(-x) = \exp(x)$. Generally, it is feasible to set $\phi(x) = \varphi(-x)$. As such, to build a new objective, one only needs to find a function that is increasing and has non-decreasing derivative. See Figure 12. In addition, all linear combinations of feasible $(\phi, \varphi)$ pairs also lie in the family.

It is worth noting that $k(f)$ is also optimized here and it is actually necessary for Property-(c) and Property-(d). This is the key difference when the Lipschitz condition is extended to general objectives. The underlying reason for the need of also minimizing $k(f)$ comes from the existence of case "$\nabla_{f^*(x)} \partial_x J_D = 0$ for $P_g(x) \neq P_r(x)$", which does not hold when the objective is Wasserstein distance. Minimizing $k(f)$ guarantees that the only Nash Equilibrium is "$P_g = P_r$ with $k(f^*) = 0$". On the other hand, if $k(f)$ is not minimized towards zero, Wasserstein distance dual form based GANs are not guaranteed to have zero gradient $\nabla_x f^*(x)$ at the convergence state $P_g = P_r$. It indicates that minimizing $k(f)$ is also beneficial to the Wasserstein GAN (Arjovsky et al., 2017).

## 3.2 Lipschitz condition connects $P_g$ and $P_r$ through $f^*(x)$

From Theorem 1, we know that for any point $x$, as long as $\partial_x J_D$ does not hold a zero gradient with respect to $f^*(x)$, $f^*(x)$ must be bounded by another point $y$ such that $|f^*(y) - f^*(x)| = k(f^*) \cdot \|x - y\|$. We here further clarify that, when there is a bounding relationship, it must involve both real sample(s) and fake sample(s). More formally, we have

**Theorem 2.** If $f^* = \arg\min_f [J_D + \lambda \cdot k(f)^2]$, then

- $\forall x \in \bar{P}_g$, if $\exists z_{\neq x} \in \bar{P}_g \cup \bar{P}_r$ such that $|f^*(x) - f^*(z)| = k(f^*) \cdot \|x - z\|$, then $\exists y_{\neq x} \in \bar{P}_r$ such that $f^*(y) - f^*(x) = k(f^*) \cdot \|x - y\|$,

- $\forall y \in \bar{P}_r$, if $\exists z_{\neq y} \in \bar{P}_g \cup \bar{P}_r$ such that $|f^*(z) - f^*(y)| = k(f^*) \cdot \|z - y\|$, then $\exists x_{\neq y} \in \bar{P}_g$ such that $f^*(y) - f^*(x) = k(f^*) \cdot \|x - y\|$.

The intuition behind the above theorem is that samples from the same distribution, e.g., the fake samples, will not bound each other. It is worth noticing that there might exist a chain of bounding relationships that involves a dozen of fake samples and real samples, and these points all lie in the same line and bounds each other.

Under the Lipschitz condition, the **bounded line** in the value surface of $f^*$ is the basic **building block** that connects $P_g$ and $P_r$, and each fake sample lies in one of the bounded lines. Next we will further interpret the implication of bounding relationship and show that it guarantees meaningful $\nabla_x f^*(x)$ for all involved points.

## 3.3 Lipschitz condition ensures the convergence of $\nabla_x f^*(x)$-based GANs

Recall that the proposition in Eq. (11) states that $\nabla_x f^*(x_t) = \frac{y-x}{\|y-x\|}$. This is actually a direct consequence of bounding relationship between $x$ and $y$. We formally state it as follows:

**Theorem 3.** Assume $f(x)$ is differentiable and $k$-Lipschitz continuous. For all $x$ and $y$ which satisfy $x \neq y$ and $f(y) - f(x) = k \cdot \|x - y\|$, we have $\nabla_x f(x_t) = k \cdot \frac{y-x}{\|y-x\|}$, where $x_t = tx + (1-t)y$ for $0 \leq t \leq 1$.

In other words, if two points $x$ and $y$ bound each other in terms of $f(y) - f(x) = k \cdot \|x - y\|$, there is a straight line between $x$ and $y$ in the value surface of $f$. Any point in this line holds the maximum gradient slope $k$, and the direction of these gradient all point towards the $x \to y$ direction. Combining Theorem 1 and Theorem 2, we can conclude that when $P_g$ and $P_r$ are disjoint, $\nabla_x f^*(x)$ for each sample $x \sim P_g$ points to a sample $y \sim P_r$, which guarantees that $P_g$ is moving towards $P_r$.

In fact, Theorem 1 provides further guarantee on the convergence. Property-(b) implies that for any $x \sim P_g$ that does not lies in $P_r$, $\nabla_x f^*(x)$ points to some real sample $y \sim P_r$. In the fully overlapped case, according to Property-(c), unless $P_g = P_r$, there exists a pair $(x, y)$ in bounding relationship and $\nabla_x f^*(x)$ pulls $x$ towards $y$. Property-(d) guarantees that the only Nash Equilibrium is "$P_g = P_r$". The proof of Theorem 3 is provided in Appendix D.

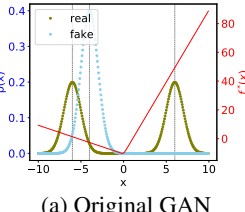
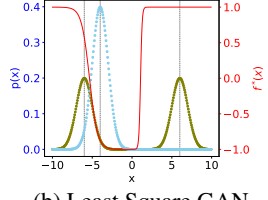
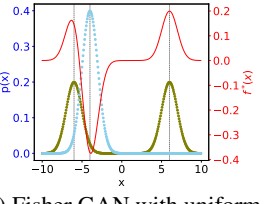

| (a) Original GAN | (b) Least Square GAN | (c) Fisher GAN with uniform $\mu$ |

Figure 2: The source of Mode Collapse. In traditional GANs, $f^*(x)$ is a function of the local densities $P_g(x)$ and $P_r(x)$. Given $f^*(x)$ is an increasing function of $P_r(x)$ and decreasing function of $P_g(x)$, when fake samples get close to a mode of the $P_r$, $\nabla_x f^*(x)$ move them towards the mode.

## 4 EXTENSIONS AND DISCUSSIONS

### 4.1 FROM DISJOINT CASE TO OVERLAPPING CASE

In Section 2, we discuss the problem of $f^*(x)$ and $\nabla_x f^*(x)$ in the case where $P_g$ and $P_r$ are disjoint. In this section, we extend our discussion to the overlapping case. In the disjoint case, we argue that "$f^*(x)$ on $P_g$ does not reflect any information about the location of other points in $P_r$" will lead to an unfeasible $\nabla_x f^*(x)$ and thus non-convergence. In the overlapping and continuous case, things are actually different, $f^*(x)$ around each point is also defined, and its gradient $\nabla_x f^*(x)$ now reflects the local variation of $f^*(x)$.

For most traditional GANs, $f^*(x)$ mainly reflects the local information about the density $P_g(x)$ and $P_r(x)$. However, it is worth noting that $f^*(x)$ is usually an increasing function with respect to $P_r(x)$ while a decreasing function with respect to $P_g(x)$. For instance, $f^*(x)$ in the original GAN is $\log P_r(x)/P_g(x)$. Optimizing the generator according $\nabla_x f^*(x)$ will move sample $x$ towards the direction of increasing $f^*(x)$. Because $f^*(x)$ positively correlates with $P_r(x)$ and negatively correlated with $P_g(x)$, it in sense means $x$ is becoming more real. However, such a local greedy strategy turns out to be a fundamental cause of mode collapse.

### 4.2 THE CAUSE OF MODE COLLAPSE: THE LOCALITY OF $f^*(x)$ AND $\nabla$

Mode collapse is a notorious problem in GANs' training, which refers to the phenomenon that the generator only learns to produce part of $P_r$. Many literatures try to study the source of mode collapse (Che et al., 2016; Metz et al., 2016; Kodali et al., 2017; Arora et al., 2017) and measure the degree of mode collapse (Odena et al., 2016; Arora & Zhang, 2017).

The most recognized cause of mode collapse is that, if the generator is much stronger than the discriminator, it may learn to only produce the sample(s) in the local or global maximum of $f(x)$ for the current discriminator. This argument is true for most of GAN models. However, from our perspective on $f^*(x)$ and its gradient, there actually exists a much more fundamental cause of mode collapse, i.e., the locality of $f^*(x)$ in traditional GANs and the locality of gradient operator $\nabla$.

In traditional GANs, $f^*(x)$ is a function of local densities $P_g(x)$ and $P_r(x)$, which is local, and the gradient operator $\nabla$ is also a local operator. As the result, $\nabla_x f^*(x)$ only reflects its local variations and cannot capture the statistic of $P_r$ and $P_g$ that is far from itself. If $f^*(x)$ in the surrounding area of $x$ is well-defined, $\nabla_x f^*(x)$ will move $x$ towards the **nearby** location where the value of $f^*(x)$ is higher. It does not take the global status into account.

The typical result is that when fake samples get close to a mode of the $P_r$, they move towards the mode and get stuck there (due to the locality). Assume $P_r$ consists of two Gaussian distributions (A and B) that are distant from each other, while the current $P_g$ is uniformly distributed over its support and close to real Gaussian A. In this case, $\nabla_x f(x)$ of all fake samples will point towards the center of Gaussian A. If $P_g$ is a Gaussian with the same standard deviation as Gaussian A, $\nabla_x f(x)$ in original GAN and Least-Square GAN shows almost identical behaviors, which is illustrated in Figure 2. In Fisher GAN, if $\mu(x)$ is uniform, the case is even worse: a large amount of points that are relatively far from Gaussian A will move away from A (but the direction is not necessarily towards B, though in our 1-D case it is). This observation again supports our argument that "a well-defined distance metric does not necessarily guarantee the convergence", and the validity of $\nabla_x f^*(x)$ is still necessary even if $P_g$ and $P_r$ is continuous and overlapped.

### 4.3 Explanation on the empirical success of traditional GANs

Though traditional GANs does not have any guarantee on its convergence, it has already achieved its great success. The reason is that having no guarantee does not mean it cannot converge. It turns out extensive parameter-tuning actually increases the probability of the convergence.

As shown in Appendix A, hyper-parameters are important in influencing the value surface of $f^*(x)$. Some typical settings (e.g., simplified neural network architecture, relu or leaky relu activation, relatively high learning rate, Adam optimizer, etc.) tend to form a relatively smooth value surface (e.g., monotonically increasing from $P_g$ to $P_r$), making $\nabla_x f^*(x)$ much more meaningful. That is, one can find these settings, where $\nabla_x f^*(x)$ or $\nabla_x f(x)$ is more favourable, to enable traditional GANs to work. In opposite, we have tried highly-nonlinear activation such as swish (Ramachandran et al., 2018) in the discriminator. It turns out traditional GANs are very likely to fail. In contrast, our proposed Lipschitz constraint based GANs are compatible with highly-nonlinear activation. Another important empirical technique is to delicately balance the generator and the discriminator or limit the capacity of the discriminator. This is to avoid the fatal optimal $f^*(x)$. All these could possibly make traditional GANs work. **However, the consequence is that these GANs are very sensitive to hyper-parameters and hard to use.**

### 4.4 The relation between Lipschitz condition and Wasserstein distance

Most literature presents the dual form of Wasserstein distance with the Lipschitz condition. However, it is worth noticing that the Lipschitz condition is actually stronger than the necessary one in the dual form of Wasserstein distance. Recall that in the dual form of Wasserstein distance, the constraint can be more compactly written as (introduced in Section 2.3 and proved in Appendix I)

$$f(x) - f(y) \leq d(x, y), \ \forall x \sim P_r, \forall y \sim P_g. \tag{13}$$

However, it is usually written as 1-Lipschitz condition, which is

$$f(x) - f(y) \leq d(x, y), \forall x, \forall y. \tag{14}$$

The key difference is that the constraint in Eq. (13) restricts the range of $x$ and $y$, but Lipschitz condition (Eq. (14)) does not have the restriction on the range, thus the latter is the sufficient condition of the former one. It is also worth noticing that, though Lipschitz condition is stronger than the compact one, it does not affect the final solution (Appendix I). In other words, Lipschitz condition is a safe extension of the compact constraint. And if the supports of $P_g$ and $P_r$ are the entire space, Eq. (13) and Eq. (14) are actually identical; in such condition, Wasserstein distance in its dual form always works. **However, $P_g$ and $P_r$ are usually disjoint in GANs**. Therefore, **using the strong Lipschitz condition is necessary** to ensure the validity of the dual form of Wasserstein distance in $\nabla_x f^*(x)$-based updating, and the constraint in Eq. (13) is not enough as shown in Section 2.3.

### 4.5 Contribution clarification

This work is substantially different from Wasserstein GAN (Arjovsky et al., 2017). Though the final solution in Wasserstein GAN is sound, its main argument is off the point. The main argument for the benefit of Wasserstein distance in (Arjovsky et al., 2017) is that it can properly measure the distance between two distributions no matter whether their supports are disjoint, i.e., Wasserstein distance is a good distance metric. However, according to our analysis in Section 2.4, a proper distance metric does not necessarily ensure the convergence of GAN; more specifically, Wasserstein distance in the dual form with compacted constraint also cannot provide meaningful gradient through $\nabla_x f^*(x)$.

In addition, we have shown that Lipschitz constraint is able to ensure the convergence of GANs in a family of GAN objectives, which is not restricted to Wasserstein distance. For example, Lipschitz constraint is also introduced to original GAN in (Miyato et al., 2018; Kodali et al., 2017) and shows improvements on the quality of generated samples. As a matter of fact, the original GAN objective $\phi(x) = \varphi(-x) = -\log(\sigma(-x))$ is another instance in our proposed family and thus our analysis explains why and how it works.

It is also worth noticing that $J_D$ in our formulation is not derived from any well-established distance metric; it is derived based on Lipschitz constraint. As we have shown that a well-established distance or divergence does not necessarily ensure the convergence, we hope our trial could shed light on the new direction of GANs.

Last but no least, though we do not discuss the generator's objective, our analysis indicates that the **minimax** in terms of $\psi$ in Eq. (1) is not essential, because it only influences the scale of the gradient. Nevertheless, the function $\psi$ does influence the updating of the generator, and we leave the detailed investigation as future work. Another example that has a theoretically meaningful $\nabla_x f^*(x)$ is Coulomb GAN (Unterthiner et al., 2017), which is also derived neither from minimax game nor from well-defined distance metric.

### 4.6 RELATED WORK

Fedus et al. (2017) also argued that divergence is not the primary guide of the training of GANs and pointed out that the gradient does not necessarily related to the divergence. However, they tended to believe that original GAN with non-saturating generator objective can somehow work. As we have proved before, given the optimal $f^*$, the original GAN has no guarantee on its convergence. And we argue that practical work scenarios benefit from parameter-tuning.

Some work study the suboptimal $f(x)$ (Mescheder et al., 2017; 2018; Arora et al., 2017), which is another important direction for understanding GANs theoretically. While the behaviors of suboptimal can be slightly different, we think the optimal $f^*(x)$ should well-behave in the first place.

Researchers also found that applying Lipschitz constraint to the generator also benefits the quality of generated samples (Zhang et al., 2018; Odena et al., 2018). In addition, researchers also investigated implementation of Lipschitz constraint in GANs (Gulrajani et al., 2017; Petzka et al., 2017; Miyato et al., 2018). However, this branch of related work is out of scope of the discussion in this paper.

## 5 EXPERIMENTS

In this section, we present the experiment results on our proposed objectives for GANs. The anonymous code is provided at `http://bit.ly/2Kvbkje`.

### 5.1 VERIFYING THE OBJECTIVE FAMILY AND ITS GRADIENT $\nabla_x f^*(x)$

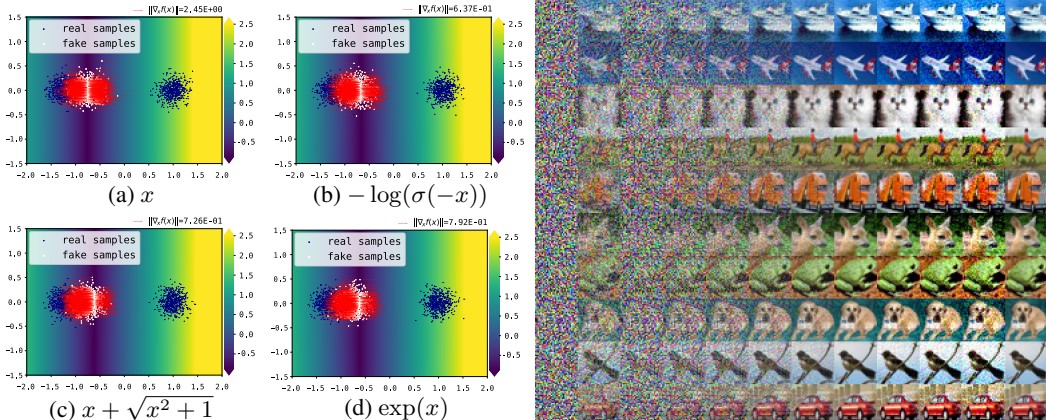

Figure 3: Verifying the objective family          Figure 4: $\nabla_x f^*(x)$ gradation with CIFAR-10

We verify a set of $\phi$ and $\varphi$ satisfying Eq. (12): (a) $\phi(x) = \varphi(-x) = x$; (b) $\phi(x) = \varphi(-x) = -\log(\sigma(-x))$; (c) $\phi(x) = \varphi(-x) = x + \sqrt{x^2 + 1}$; (d) $\phi(x) = \varphi(-x) = \exp(x)$. As shown in Figure 3, the gradient of each generated sample is towards a real sample.

We further verify $\nabla_x f^*(x)$ with the real-world data, using **ten CIFAR-10 images** as $P_r$ and **ten noise images** as $P_g$ to make the solving of $f^*(x)$ feasible. The result is shown in Figure 4, where The leftmost in each row are the $x \sim P_g$ and the second are their gradient $\nabla_x f(x)$. The interior are $x + \epsilon \cdot \nabla_x f(x)$ with increasing $\epsilon$, which will pass through a real sample, and the rightmost are the nearest $y \sim P_r$. This result visually demonstrates that the gradient of a generated sample is towards the direction of one real sample. Note that the final results of this experiment keep almost identical when varying the loss metric $\phi(x)$ and $\varphi(x)$ in the family.

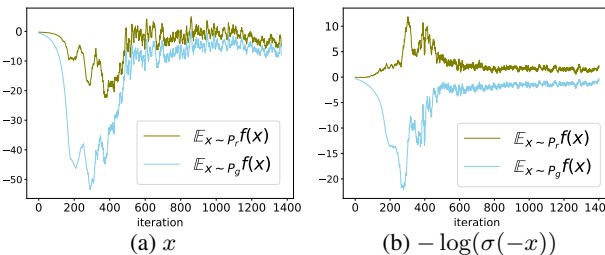
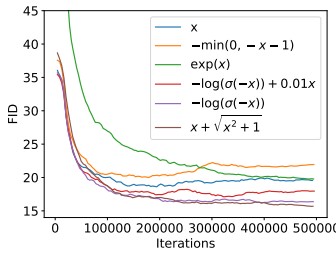

Figure 5: $f^*(x)$ in new objective is more stable.  Figure 6: Training curves on CIFAR-10.

## 5.2 STABILIZING $f^*(x)$ WITH NEW OBJECTIVES

Wasserstein distance is a special case in our proposed family of objectives where $\phi''(x) = \varphi''(x) = 0$. As a result, $f^*(x)$ under the Wasserstein distance objective where $\phi(x) = \varphi(-x) = x$ has a free offset, which means given a $f^*(x)$, $f^*(x) + b$ with any $b \in \mathbb{R}$ is also an optimal. In practice, this behaves as an oscillatory $f(x)$ during training. Any other instance of our new proposed objectives does not have this problem. We illustrate this practical difference in Figure 5.

## 5.3 BENCHMARK ON UNSUPERVISED IMAGE GENERATION TASKS

Table 2: Quantitative comparisons on unsupervised image generation tasks.

| Objective | CIFAR-10 | | Tiny ImageNet | | Oxford 102 Flower | |
|---|---|---|---|---|---|---|
| | FID | IS | FID | IS | FID* | IS* |
| $-\min(0, -x - 1)$ | $21.58 \pm 0.21$ | $7.43 \pm 0.04$ | $16.22 \pm 0.33$ | $\mathbf{8.58 \pm 0.08}$ | $9.72 \pm 0.51$ | $21.91 \pm 0.18$ |
| $x$ | $19.64 \pm 0.23$ | $7.66 \pm 0.03$ | $18.81 \pm 0.58$ | $8.20 \pm 0.05$ | $9.74 \pm 0.63$ | $21.66 \pm 0.22$ |
| $-\log(\sigma(-x))$ | $16.36 \pm 0.09$ | $\mathbf{8.49 \pm 0.11}$ | $\mathbf{15.94 \pm 0.33}$ | $8.42 \pm 0.04$ | $9.40 \pm 0.49$ | $21.82 \pm 0.11$ |
| $x + \sqrt{x^2 + 1}$ | $\mathbf{15.76 \pm 0.13}$ | $8.04 \pm 0.04$ | $16.83 \pm 0.41$ | $8.35 \pm 0.09$ | $\mathbf{9.16 \pm 0.52}$ | $\mathbf{21.96 \pm 0.19}$ |
| $\exp(x)$ | $19.82 \pm 0.13$ | $7.79 \pm 0.03$ | $20.45 \pm 0.15$ | $8.06 \pm 0.05$ | $9.90 \pm 0.72$ | $21.91 \pm 0.22$ |
| $-\log(\sigma(-x)) + 0.01x$ | $18.32 \pm 0.15$ | $7.75 \pm 0.04$ | $16.09 \pm 0.23$ | $8.47 \pm 0.10$ | $9.50 \pm 0.39$ | $21.91 \pm 0.20$ |

Finally, we fix $\psi(x) = -x$ in the generator's objective and compare various objectives on unsupervised image generation tasks. The results of Inception Score (Salimans et al., 2016) and Frechet Inception Distance (Heusel et al., 2017b) are presented in Table 2. We also include the hinge loss $\phi(x) = \varphi(-x) = -\min(0, -x - 1)$ which used in (Miyato et al., 2018). We use a classifier on Oxford 102 Flower Dataset for the evaluation of FID and Inception Score for results on Oxford 102.

The gradient of $\exp(x)$ varies significantly and we find it requires a small learning rate to avoid explosion. The objectives $x + \sqrt{x^2 + 1}$ and $-\log(\sigma(-x))$ achieve the best performances. This is probably because they have bounded gradient and reduce the gradient of well-identified points towards zero, which enables the discriminator to pay more attention to these ill-identified. Hinge loss $-\min(0, -x - 1)$ does not lie in our proposed objective family and turns out to be unstable and performs unsatisfactory in same cases. We also plot the training curve in terms of FID in Figure 6.

Due to page limitation, we leave the details, visual results and more experiments in the Appendix.

## 6 CONCLUSION

In this paper we have shown that the fundamental cause of failure in training of GANs stems from the unreliable $\nabla_x f^*(x)$. Specifically, when $P_g$ and $P_r$ are disjoint, $\nabla_x f^*(x)$ for fake sample $x \sim P_g$ tells nothing about $P_r$, making it impossible for $P_g$ to converge to $P_r$. We have further demonstrated that even Wasserstein distance in a more compact dual form (still is equivalent to Wasserstein distance and can properly measure the distance between distributions) also suffers from the same problem when $P_g$ and $P_r$ are disjoint. This implies that "whether a distance metric can properly measure the distance" does not yet touch the key of non-convergence of GANs. We have highlighted in this paper that a well-defined distance metric does not necessarily guarantee the convergence of GANs because its $\nabla_x f^*(x)$ can be meaningless. Therefore, if we update the generator based on $\nabla_x f^*(x)$, we need to pay more attention on the design of $f^*(x)$. Furthermore, to address the aforementioned problem, we have proposed the Lipschitz-continuity condition as a general solution to make $\nabla_x f^*(x)$ reliable and ensure the convergence of GANs, which works well with a large family of GAN objectives. In addition, we have shown that in the overlapping case, $\nabla_x f^*(x)$ is also problematic which turns out to be an intrinsic cause of mode collapse in traditional GANs.

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

## A EXPERIMENTS: THE INFLUENCE OF HYPER-PARAMETERS

The value surface of traditional GANs is highly depended on the network and training hyper-parameters. We here plot the value surface of Least-Square GAN with various hyper-parameter settings, to give directly impression on how these parameters influence GANs training. Not very strictly, but our empirical code is: (i) a low-capacity network tends to learn a simple surface; (ii) SGD tends to learn a more complex surface than ADAM; (iii) large learning rate tends to learns a simpler surface than small learning rate; (iv) highly nonlinear activation function tends to result in more complex value surface.

Though hyper-parameters tuning could possibly make traditional GANs work, it also makes these GANs hard to use, sensitive to hyper-parameters and easily broken.

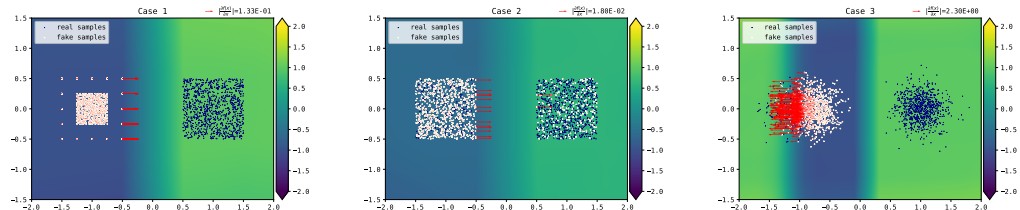

Figure 7: ADAM optimizer with lr=1e-2, beta1=0.0, beta2=0.9. MLP with RELU activations, #hidden units=1024, #layers=1.

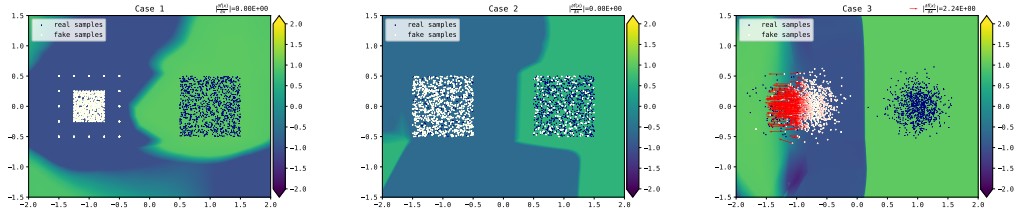

Figure 8: ADAM optimizer with lr=1e-2, beta1=0.0, beta2=0.9. MLP with RELU activations, #hidden units=1024, #layers=4.

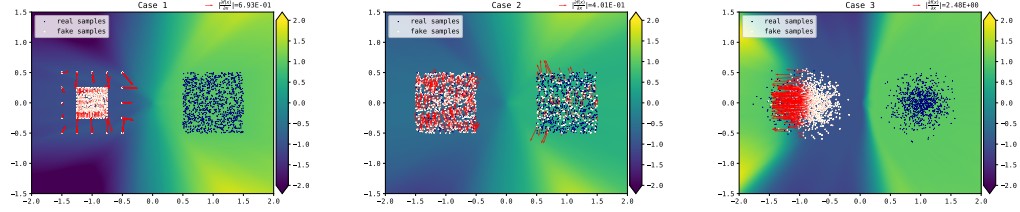

Figure 9: ADAM optimizer with lr=1e-5, beta1=0.0, beta2=0.9. MLP with RELU activations, #hidden units=1024, #layers=4.

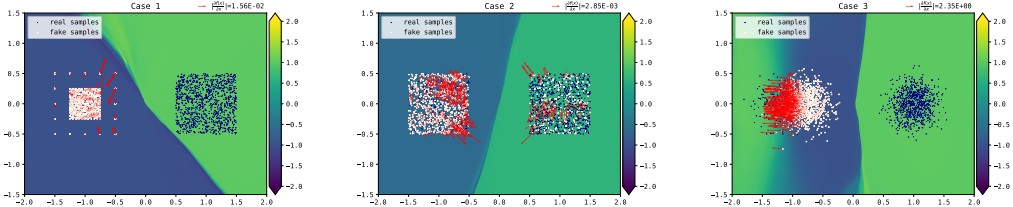

Figure 10: SGD optimizer with lr=1e-3. MLP with SELU activations, #hidden units=128, #layers=64.



Figure 11: SGD optimizer with lr=1e-4. MLP with SELU activations, #hidden units=128, #layers=64.

# B  VARIOUS $\phi(x)$ AND $\varphi(x)$ THAT SATISFIES EQ. 12

For Lipschitz constraint based GANs, $\phi(x)$ and $\varphi(x)$ are required to satisfy Eq. 12. Eq. (12) is actually quite general and there exists many other instances, e.g., $\phi(x) = \varphi(-x) = x$, $\phi(x) = \varphi(-x) = -\log(\sigma(-x))$, $\phi(x) = \varphi(-x) = x + \sqrt{x^2 + 1}$, $\phi(x) = \varphi(-x) = \exp(x)$, etc. We plot these instances of $\phi(x)$ and $\varphi(x)$ in Figure 12.

Generally, it is feasible to set $\phi(x) = \varphi(-x)$. Note that rescaling and offsetting along the axes are trivial operation to found more $\phi(x)$ and $\varphi(x)$ within a function class, and linear combination of two or more $\phi(x)$ or $\varphi(x)$ from different function classes also keep satisfying Eq. 12.

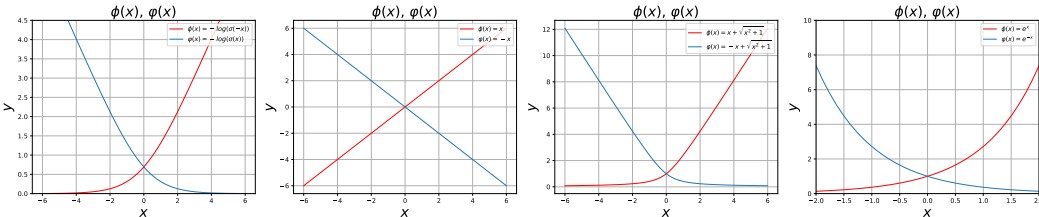

Figure 12: Various $\phi(x)$ and $\varphi(x)$ that satisfies Eq. 12.

# C  GENERATED IMAGES AND TRAINING CURVES

Training curves on Tiny ImageNet are plotted in Figure 13. And comparisons on the visual results among different objectives are also provided in Figure 14, Figure 15 and Figure 16.

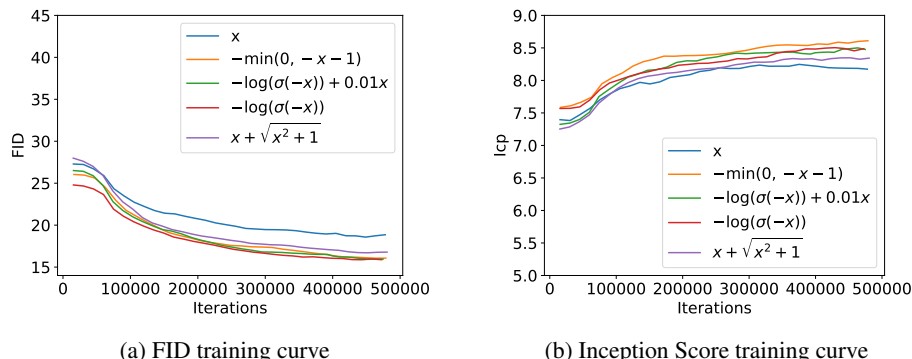

(a) FID training curve

(b) Inception Score training curve

Figure 13: FID and ICP (Inception Score) training curves of different objectives on Tiny ImageNet.

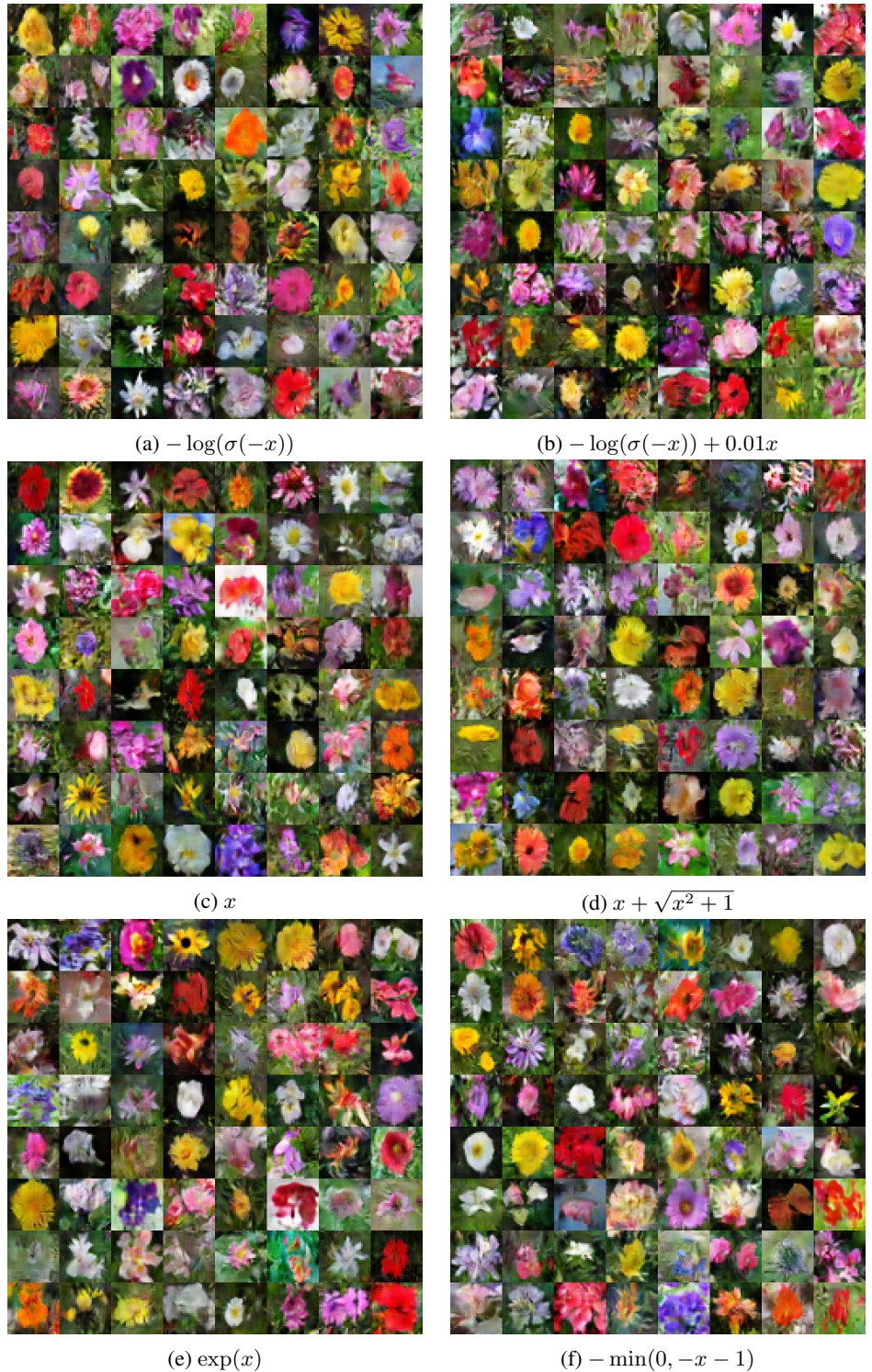

Figure 14: Random Samples of Lipschitz GAN trained of different objectives on Oxford 102.

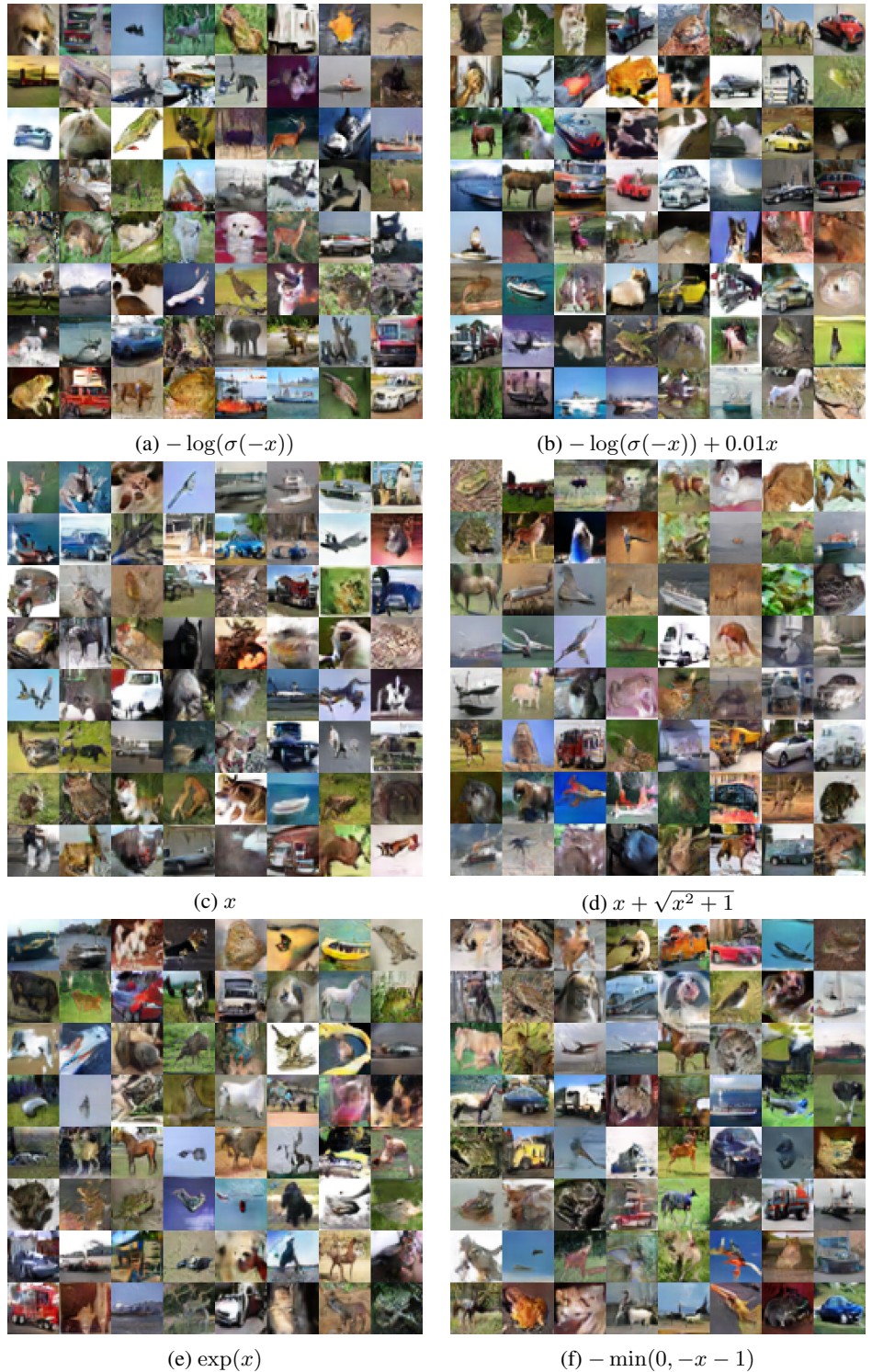

(a) $-\log(\sigma(-x))$

(b) $-\log(\sigma(-x)) + 0.01x$

(c) $x$

(d) $x + \sqrt{x^2 + 1}$

(e) $\exp(x)$

(f) $-\min(0, -x - 1)$

Figure 15: Random Samples of Lipschitz GAN trained of different objectives on Cifar-10.

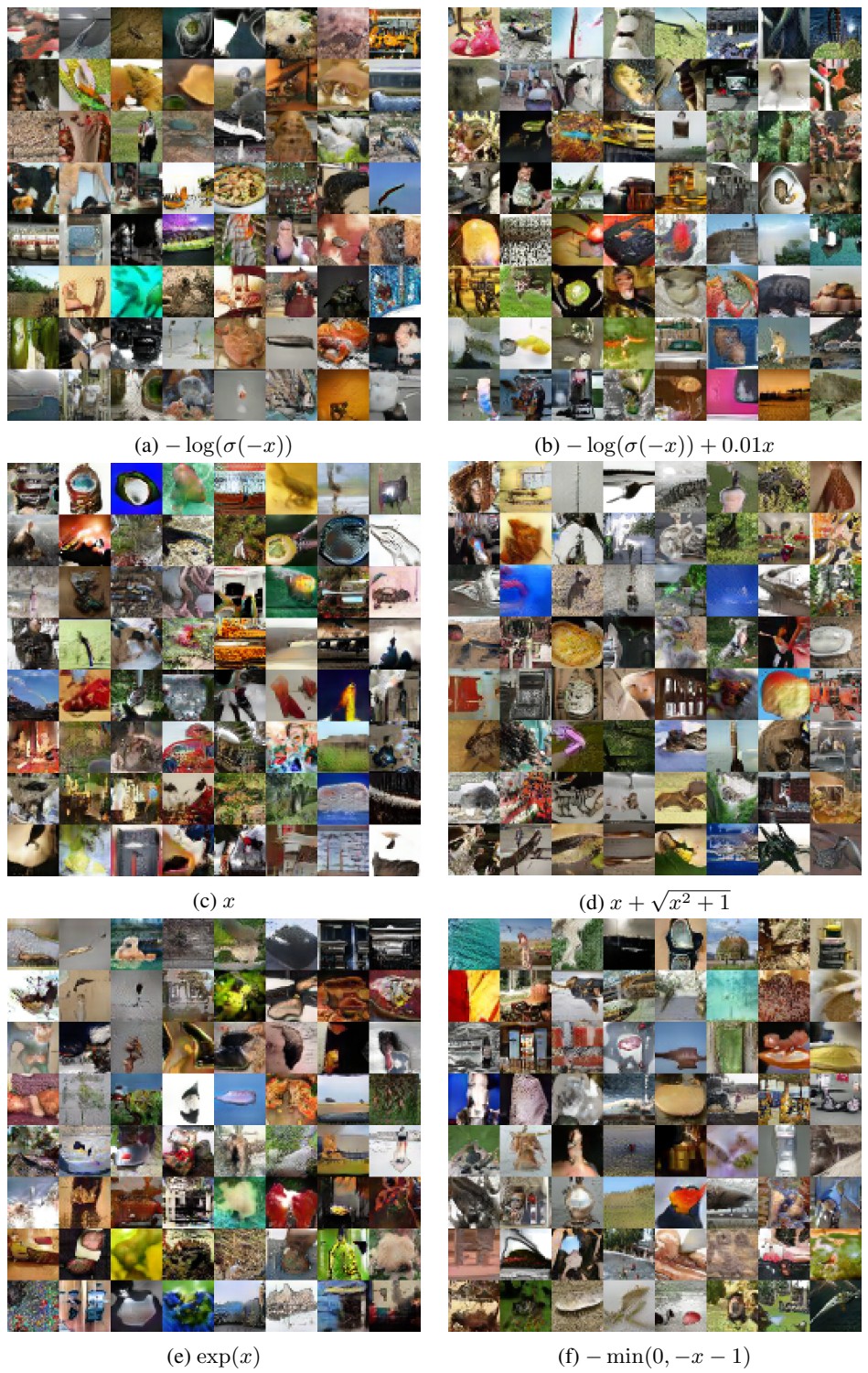

(a) $-\log(\sigma(-x))$

(b) $-\log(\sigma(-x)) + 0.01x$

(c) $x$

(d) $x + \sqrt{x^2 + 1}$

(e) $\exp(x)$

(f) $-\min(0, -x - 1)$

Figure 16: Random Samples of Lipschitz GAN trained of different objectives on Tiny Imagenet.

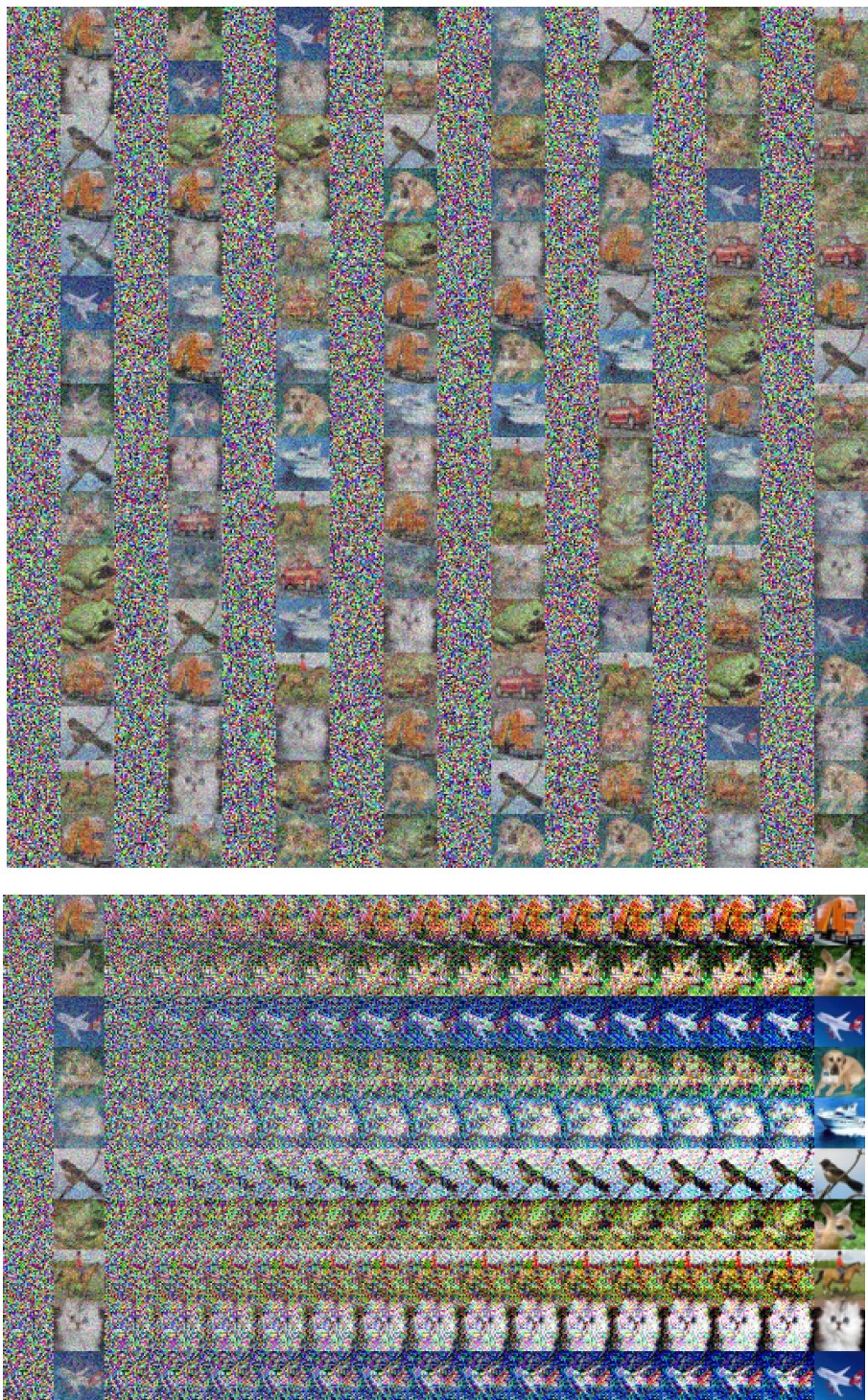

Figure 17: The gradient of Lipschitz constraint based GANs with real world data, where $P_r$ consists of ten images and $P_g$ is Gaussian noise. Up: Each odd column are $x \sim P_g$ and the nearby column are their gradient $\nabla_x f^*(x)$. Down: the leftmost in each row are $x \sim P_g$, the second are their gradients $\nabla_x f^*(x)$, the interior are $x + \epsilon \cdot \nabla_x f^*(x)$ with increasing $\epsilon$, and the rightmost are the nearest $y \sim P_r$.

## D    PROOF OF THEOREM 3 AND THE NECESSITY OF EUCLIDEAN DISTANCE

In this section, we delve deeply into the relationship between gradient properties and different norms in Lipschitz-continuity condition. We will prove Theorem 3, *i.e.* Lipschitz continuity with $l_2$-norm (Euclidean Distance) can guarantee the gradient direction of $\nabla f^*(x)$, and at the same time, demonstrate that the other norms do not have this property. To start with, we give the proof of Theorem 3 in the following.

*Proof.*

Let $(x, y)$ be such that $x \neq y$, and we define $x_t = x + t \cdot (y - x)$ with $t \in [0, 1]$. We claim that: if $f(x)$ is k-Lipschitz with respect to $\|.\|_p$ and $f(y) - f(x) = k\|x - y\|_p$, then $f(x_t) = f(x) + t \cdot k\|x - y\|_p$.

As we know $f(x)$ is k-Lipschitz, with the property of norms, we have

$$
\begin{aligned}
f(y) - f(x) &= f(y) - f(x_t) + f(x_t) - f(x) \\
&\leq f(y) - f(x_t) + k\|x_t - x\|_p = f(y) - f(x_t) + t \cdot k\|x - y\|_p \\
&\leq k\|y - x_t\|_p + t \cdot k\|x - y\|_p = k \cdot (1 - t)\|x - y\|_p + t \cdot k\|x - y\|_p \\
&= k\|x - y\|_p.
\end{aligned}
\tag{15}
$$

Given $f(x) - f(y) = k\|x - y\|_p$, it implies all the inequalities need to be equalities. Therefore, $f(x_t) = f(x) + t \cdot k\|x - y\|_p$.

It is clear that: given $f(x)$ is k-Lipschitz with respect to $\|.\|_2$, if $f(x)$ is differentiable at $x_t$, then $\|\nabla f(x_t)\|_2 \leq k$. With $f(x_t) = f(x) + t \cdot k\|x - y\|_2$, the directional derivative of $f(x)$ on the direction $v = \frac{y-x}{\|y-x\|_2}$ at $x_t$ is equal to $k$,

$$
\begin{aligned}
\frac{\partial f(x_t)}{\partial v} &= \lim_{h \to 0} \frac{f(x_t + hv) - f(x_t)}{h} = \lim_{h \to 0} \frac{f(x_t + h\frac{y-x}{\|y-x\|_2}) - f(x_t)}{h} \\
&= \lim_{h \to 0} \frac{f(x_{t+\frac{h}{\|y-x\|_2}}) - f(x_t)}{h} = \lim_{h \to 0} \frac{\frac{h}{\|y-x\|_2} \cdot k\|y - x\|_2}{h} = k.
\end{aligned}
\tag{16}
$$

Note that $\|v\|_2 = \|\frac{y-x}{\|y-x\|_2}\|_2 = 1$, *i.e.* $v$ is a unit vector. Now,

$$
k^2 = k\frac{\partial f(x_t)}{\partial v} = k\langle v, \nabla f(x_t)\rangle = \langle kv, \nabla f(x_t)\rangle \leq \|kv\|_2\|\nabla f(x_t)\|_2 = k^2.
\tag{17}
$$

As the equality holds only when $\nabla f(x_t) = kv = k\frac{y-x}{\|y-x\|_2}$, we prove that $\nabla f(x_t) = k\frac{y-x}{\|y-x\|_2}$.    $\square$

Above proof utilizes the property that $\|\nabla f(x_t)\|_2 \leq k$, which is derived from that $f(x)$ is k-Lipschitz with respect to $\|.\|_2$. However, other norms do not hold this property. Specifically, according to the theory in (Shalev-Shwartz et al., 2012): if a convex and differentiable function $f$ is k-Lipschitz over $\mathcal{S}$ with respect to norm $\|.\|_p$, then the Lipschitz continuity actually implies a bound on the dual norm of gradients, *i.e.* $\|\nabla f\|_q \leq k$. Here $\|.\|_q$ is the dual norm of $\|.\|_p$, which satisfies the equation that $\frac{1}{p} + \frac{1}{q} = 1$. As we could notice, a norm is equal to its dual norm if and only if $p = 2$. Switching to $l_p$-norm with $p \neq 2$, it is actually bounding the $l_q$-norm of the gradients. However, bounding the $l_q$-norm of the gradients does not guarantee the gradient direction at fake samples point towards real samples. A counter-example is provided as follows.

Consider a function $g(x, y) = x + y$ on $\mathbb{R}^2$. $\forall\, p_1 = (x_1, y_1), p_2 = (x_2, y_2)$, there is $g(p_1) - g(p_2) = g(x_1, y_1) - g(x_2, y_2) = (x_1 - x_2) + (y_1 - y_2) \leq |x_1 - x_2| + |y_1 - y_2| = \|p_1 - p_2\|_1$, which means $g$ is a 1-Lipschitz function with respect to $l_1$-norm. According to above analysis, the dual norm of $\nabla g$ is bounded, *i.e.* $\|\nabla g\|_\infty \leq 1$. Actually $\nabla g$ is equal to $(1, 1)$ at every point in $\mathbb{R}^2$ with $\|\nabla g\|_\infty = 1$. Selecting two points $A = (0, 0)$ and $B = (2, 1)$, we have $g(A) - g(B) = \|A - B\|_1$, however, $\nabla g(A) = (1, 1)$ is not pointing towards $B$.

# E    ON THE IMPLEMENTATION OF K-LIPSCHITZ FOR GANS

Typical techniques for enforcing k-Lipschitz includes: spectral normalization (Miyato et al., 2018), gradient penalty (Gulrajani et al., 2017), and Lipschitz penalty (Petzka et al., 2017). Before moving into the detailed discussion of these methods, we would provide several important notes in the first place.

**Firstly**, enforcing k-Lipschitz in the blending-region of $P_g$ and $P_r$ is actually sufficient. Define $B(\mu, \nu) = \{\hat{x} = x \cdot t + y \cdot (1 - t) \mid x \sim \mu \wedge y \sim \nu \wedge t \in [0, 1]\}$. It is clear that $f(x)$ is 1-Lipschitz in $B(\mu, \nu)$ implies $f(x) - f(y) \leq d(x, y), \forall x \in \mu, \forall y \in \nu$. Thus, it is a sufficient constraint for Wasserstein distance in Eq. 8. In fact, $f(x)$ is k-Lipschitz in $B(P_g, P_r)$ is also a sufficient condition for all properties described in Lipschitz constraint based GANs (Section 3).

**Secondly**, enforcing k-Lipschitz with regularization would provide a dynamic Lipschitz constant $k$.

**Theorem 4.**    With Wasserstein GAN objective, we have $\min_{f \in \mathcal{F}_{\text{k-Lip}}} J_D(f) = k \cdot \min_{f \in \mathcal{F}_{\text{1-Lip}}} J_D(f)$.

Assuming we know and can control the Lipschitz constant $k$ of $f(x)$, by introducing a loss, saying square loss, on $k$ respecting to a constant $k_0$, the total loss of the discriminator (critic) becomes $J(k) \triangleq \min_{f \in \mathcal{F}_{\text{k-Lip}}} J_D(f) + \lambda \cdot (k - k_0)^2$. With Lemma 4, let $\alpha = -\min_{f \in \mathcal{F}_{\text{1-Lip}}} J_D(f)$, then $J(k) = -k \cdot \alpha + \lambda \cdot (k - k_0)^2$, and $J(k)$ achieves its minimum when $k = \frac{\alpha}{2\lambda} + k_0$. When $\alpha$ goes to zero, *i.e.* $P_g$ converges to $P_r$, the optimal $k$ decreases. And when $P_g = P_r$, we have $\alpha = 0$ and optimal $k = k_0$. We choose $k_0 = 0$ in our experiments. The similar analysis applies to Lipschitz constraint based GANs and we use $\lambda \cdot k^2$ to enforcing k-Lipschitz for general Lipschitz constraint based GANs.

**For practical methods**, though spectral normalization (Miyato et al., 2018) recently demonstrates their excellent results in training GANs, spectral normalization is an absolute constraint for Lipschitz over the entire space, i.e., constricting the maximum gradient of the entire space, which is unnecessary. On the other side, we also notice both penalty methods proposed in (Gulrajani et al., 2017) and (Petzka et al., 2017) are not the exactly implementing the Lipschitz continuity condition, because it does not simply penalty the maximum gradient, but penalties all gradients towards 1, or penalties all these greater than one towards 1.

We found in our experiments that the existing methods including spectral normalization (Miyato et al., 2018), gradient penalty (Gulrajani et al., 2017), and Lipschitz penalty (Petzka et al., 2017) all fail to converge to the optimal $f^*(x)$ in many of our synthetic experiments. We thus developed a new method for enforcing k-Lipschitz and we found in our experiments that the new method stably converges to the optimal $f^*(x)$.

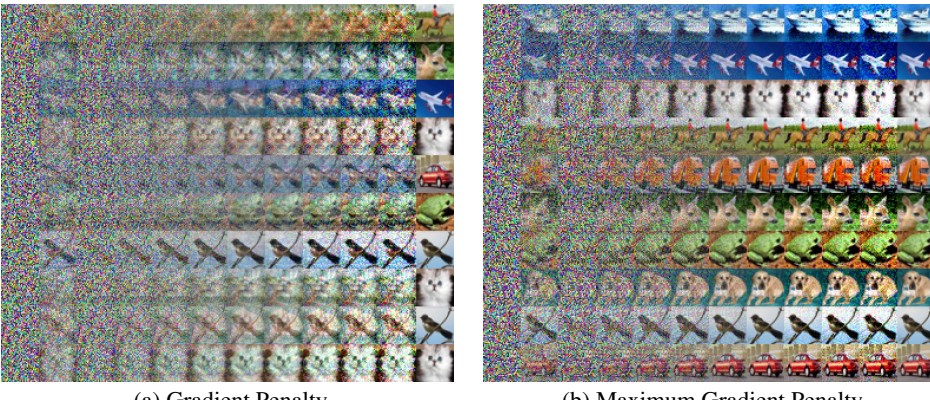

(a) Gradient Penalty                    (b) Maximum Gradient Penalty

Figure 18: Comparison between gradient penalty and maximum gradient penalty, with $P_r$ and $P_g$ consist of **ten real and noise images**, respectively. The leftmost in each row is a $x \sim P_g$ and the second is its gradient $\nabla_x f^*(x)$. The interior are $x + \epsilon \cdot \nabla_x f^*(x)$ with increasing $\epsilon$, which will pass through a real sample, and the rightmost is the corresponding $y \sim P_r$.

**The new method.** Note that the practical methods of imposing k-Lipschitz is not the key contribution of this work, and it is far from well-validated. We plan a further work on this topic for a more rigorous study. But for the necessity for understanding our paper and reproducing of experiments, we introduce it as follows.

Combining the idea of spectral normalization and gradient penalty, we developed an new regularization for Lipschitz continuity in our experiments. Spectral normalization is actually constraining the maximum gradient over the entire space. And as we argued previously, enforcing Lipschitz continuity in the blending region is sufficient. Therefore, we propose to restricting the maximum gradient over the blending region:

$$J_{\mathrm{maxgp}} = \max_{\hat{x} \sim B(\mu, \nu)} [\left\| \nabla f(x) \right\|_2^2] \tag{18}$$

In practice, we sample $\hat{x}$ from training batch as in (Gulrajani et al., 2017; Petzka et al., 2017). To improve the stability and reduce the biased introduced via batch sampling, we propose the keep track $\hat{x}$ with the maximum $\left\| \nabla f(x) \right\|_2$. A practical and light weight method is to maintain a list $S_{\mathrm{max}}$ that has the currently highest (top-k) $\left\| \nabla f(x) \right\|_2$ (initialized with random $\hat{x}$ samples), using the $S_{\mathrm{max}}$ as part of the batch estimation of $J_{\mathrm{maxgp}}$, and update the $S_{\mathrm{max}}$ after each batch updating of the discriminator. In our experiment, $S_{\mathrm{max}}$ takes 1/2 batch, and the remaining 1/2 batch are random sampled. $S_{\mathrm{max}}$ always keeps track of the maximal 1/2 samples in the batch.

We compare the practical result of gradient penalty $\mathbb{E}_{\hat{x} \sim B}[\left\| \nabla f(x) \right\|_2^2]$ and the proposed maximum gradient penalty in Figure 18. Before switching to maximum gradient penalty, we struggled for a long time and cannot achieve a high quality result as showed in Figure 18b. The other forms of gradient penalty (Gulrajani et al., 2017; Petzka et al., 2017) perform similar as $\mathbb{E}_{\hat{x} \sim B}[\left\| \nabla f(x) \right\|_2^2]$.

# F  DISCUSSION ON NO-DIFFERENTIABLE $f^*(x)$

If $f^*(x)$ is k-Lipschitz and $f^*(y) - f^*(x) = k \cdot d(x, y)$, we say that $(x, y)$ are coupled. When a sample $x$ is coupled with more than one $y$ and these $y$ lie in different directions of $x$, $f^*(x)$ is non-differentiable at $x$ and it will has sub-gradient along each direction.

When the $f^*(x)$ non-differentiable, due to the smoothness of practically-used neural network, as we noticed in the experiments, it usually behaviors as that the gradient direction is pointing in the middle of these sub-gradient (more strictly, a linear combination of these sub-gradients).

It seems that when the $P_g$ is discrete (simulating discrete token generation, such as language and music), it is easy to become non-differentiable: in the optimal transport perspective, once it is required to move to more than one targets, $f^*(x)$ is non-differentiable at this point.

One way to alleviate this above problem is adding noise (e.g. Gaussian) to each discrete token from $P_g$. The discrete token with different noises now disperse to different targets. In the practical generator for continuous token, such as images, this kind of non-differentiable problem naturally get solved.

The more serious non-differentiable problem traces back to the Monge problem (Villani, 2008), which theoretically discussed under which condition the optimal transport is a one-one mapping, which by nature solve the non-differentiable problem, as each sample now has a single target.

However, for the Monge problem is solvable, *i.e.* the mapping from $P_g$ and $P_r$ is one-one, it requires the $d(x, y)$ to be a strictly convex and super-linear (Villani, 2008). Unfortunately, the Euclidean distance, which is necessary to ensure the gradient direction from fake sample directly points toward real sample, does not fit this condition. So we currently does not figure out a practical solution to take advantage of the Monge problem related theories.

Nonetheless, even if $f^*(x)$ is non-differentiable, the gradient is also usually somehow pointing towards the real samples. And the empirical founding is that: when the $P_g$ get close to $P_r$, the non-differentiable problem diminishes.

# G    PROOF OF THE THEOREM 1

Let $J_D = \mathbb{E}_{x \sim P_g}[\phi(f(x))] + \mathbb{E}_{x \sim P_r}[\varphi(f(x))] = \int P_g(x)\phi(f(x)) + P_r(x)\varphi(f(x))dx$. Let $\partial_x J_D$ denotes $P_g(x)\phi(f(x)) + P_r(x)\varphi(f(x))$. It has $J_D = \int \partial_x J_D dx$.

Define $J = J_D + \lambda \cdot k(f)^2$, where $k(f)$ is Lipschitz constant of $f(x)$. Let $f^*(x) = \arg\min_f[J_D + \lambda \cdot k(f)^2]$. Let $J_D^*(k) = \min_{f \in \mathcal{F}_{k\text{-Lip}}} J_D = \min_{f \in \mathcal{F}_{1\text{-Lip}}, b} \mathbb{E}_{x \sim P_g}[\phi(k \cdot f(x)+b)] + \mathbb{E}_{x \sim P_r}[\varphi(k \cdot f(x)+b)]$.

**Lemma 1.**    $\forall x, \frac{\partial[\partial_x J_D]}{\partial f^*(x)} = 0$ if and only if $k(f^*) = 0$.

*Proof.*

(i) $\forall x, \frac{\partial[\partial_x J_D]}{\partial f^*(x)} = 0$ implies $k(f^*) = 0$.

For the optimal $f^*(x)$, it holds that $\frac{\partial J}{\partial k(f^*)} = \frac{\partial J_D^*}{\partial k(f^*)} + 2\lambda \cdot k(f^*) = 0$. $\forall x, \frac{\partial[\partial_x J_D]}{\partial f^*(x)} = 0$ implies $\frac{\partial J_D^*}{\partial k(f^*)} = 0$. We thus conclude that $k(f^*) = 0$.

(ii) $k(f^*) = 0$ implies $\forall x, \frac{\partial[\partial_x J_D]}{\partial f^*(x)} = 0$.

For the optimal $f^*(x)$, it holds that $\frac{\partial J}{\partial k(f^*)} = \frac{\partial J_D^*}{\partial k(f^*)} + 2\lambda \cdot k(f^*) = 0$. So $k(f^*) = 0$ implies $\frac{\partial J_D^*}{\partial k(f^*)} = 0$. $k(f^*) = 0$ also implies $\forall x, y, f^*(x) = f^*(y)$. If there exists some point $x$ such that $\frac{\partial[\partial_x J_D]}{\partial f^*(x)} \neq 0$, then, given $\forall x, y, f^*(x) = f^*(y)$, it is obviously that $\frac{\partial J_D^*}{\partial k(f^*)} \neq 0$. It is contradictory to $\frac{\partial J_D^*}{\partial k(f^*)} = 0$. Thus we has $\forall x, \frac{\partial[\partial_x J_D]}{\partial f^*(x)} = 0$.    □

**Lemma 2.**    If $\forall x, y, f^*(x) = f^*(y)$, then $P_g = P_r$.

*Proof.* $\forall x, y, f^*(x) = f^*(y)$ implies $k(f^*) = 0$. According to Lemma 1, $\forall x, \frac{\partial[\partial_x J_D]}{\partial f^*(x)} = P_g(x)\frac{\partial\phi(f^*(x))}{\partial f^*(x)} + P_r(x)\frac{\partial\varphi(f^*(x))}{\partial f^*(x)} = 0$. So $\frac{P_g(x)}{P_r(x)} = -\frac{\frac{\partial\varphi(f^*(x))}{\partial f^*(x)}}{\frac{\partial\phi(f^*(x))}{\partial f^*(x)}}$, and thus $\frac{P_g(x)}{P_r(x)}$ has a constant value, which straightforwardly implies $P_g = P_r$.    □

*Proof of Theorem 1.*

(i) Considering the $f^*(x)$, $\forall x \in \bar{P}_g \cup \bar{P}_r$, if there does not exist a $y$ such that $|f^*(y) - f^*(x)| = k(f^*) \cdot d(x, y)$, because $f^*(x)$ is the optimal, it must hold that $\frac{\partial[\partial_x J_D]}{\partial f^*(x)} = 0$. [4]

(ii) For $x \in \bar{P}_g \cup \bar{P}_r - \bar{P}_g \cap \bar{P}_r$, assuming $P_g(x) \neq 0$ and $P_r(x) = 0$, we have $\frac{\partial[\partial_x J_D]}{\partial f^*(x)} = P_g(x)\frac{\partial\phi(f^*(x))}{\partial f^*(x)} + P_r(x)\frac{\partial\varphi(f^*(x))}{\partial f^*(x)} = P_g(x)\frac{\partial\phi(f^*(x))}{\partial f^*(x)} > 0$, because $P_g(x) > 0$ and $\frac{\partial\phi(f^*(x))}{\partial f^*(x)} > 0$. Then, according to (i), there must exist a $y$ such that $|f^*(y) - f^*(x)| = k(f^*) \cdot d(x, y)$. The other situation can be proved in the same way.

(iii) According to Lemma 2, in this situation that $P_g \neq P_r$, for the optimal $f^*(x)$, there must exist at least one pair of points $x$ and $y$ such that $x \neq y$ and $f^*(x) \neq f^*(y)$. If there are no $x$ and $y$ satisfying that $|f^*(y) - f^*(x)| = k(f^*) \cdot d(x, y)$, it will be contradictory to that $f^*(x)$ is optimal, because we can construct a better $f^*$ by decreasing the value of $k(f)$ until there are two points, *e.g.* $x$ and $y$, constrained by Lipschitz constraint, *i.e.* $|f^*(y) - f^*(x)| = k(f^*) \cdot d(x, y)$.

(iv) In Nash Equilibrium state, it holds that, for any $x \in \bar{P}_g \cup \bar{P}_r$, $\frac{\partial J}{\partial k(f)} = \frac{\partial J_D^*}{\partial k(f)} + 2\lambda \cdot k(f) = 0$ and $\frac{\partial[\partial_x J_D]}{\partial f(x)}\frac{\partial f(x)}{\partial x} = 0$. We claim that in the Nash Equilibrium state, the Lipschitz constant $k(f)$ must be 0. If $k(f) \neq 0$, according to Lemma 1, there must exist a point $\hat{x}$ such that $\frac{\partial[\partial_{\hat{x}} J_D]}{\partial f(\hat{x})} \neq 0$. And according to (i), it must hold that $\exists \hat{y}$ fitting $|f(\hat{y}) - f(\hat{x})| = k(f) \cdot d(\hat{x}, \hat{y})$. According to Theorem

---

[4]Otherwise, as $f^*(x)$ is not constrained by the Lipschitz constraint, we can construct a better $f^*$ by adjusting the value of $f^*(x)$ at $x$ according to the non-zero gradient.

3, we have $\left\|\frac{\partial f(x)}{\partial x}\big|_{x=\hat{x}}\right\|_2 = k(f) \neq 0$. This is contradictory to that $\frac{\partial [\partial_x J_D]}{\partial f(x)} \frac{\partial f(x)}{\partial x}\big|_{x=\hat{x}} = 0$. Thus $k(f) = 0$, that is, $\forall x \in \bar{P}_g \cup \bar{P}_r$, $\frac{\partial f(x)}{\partial x} = 0$, which means $\forall x, y, f(x) = f(y)$. According to Lemma 2, $\forall x, y, f(x) = f(y)$ implies $P_g = P_r$. Thus $P_g = P_r$ is the only Nash Equilibrium of our system. $\qquad\square$

**Remark:** For the Wasserstein distance, $\frac{\partial [\partial_x J_D]}{\partial f^*(x)} = 0$ if and only if $P_g(x) = P_r(x)$. For the Wasserstein distance, penalizing the Lipschitz constant also benefits: at the convergence state, it holds $\frac{\partial f^*(x)}{\partial x} = 0$.

## H    ON THE IMPORTANCE OF EQ. 12

Requiring $\phi(x)$ and $\varphi(x)$ to satisfy Eq. 12 is important, because it is the non-trivial condition that makes sure $\mathbb{E}_{x \sim P_g}[\phi(f(x))] + \mathbb{E}_{x \sim P_r}[\varphi(f(x))] + \lambda \cdot k(f)^2$ has attainable global minimum with respect to $f$.

**Theorem 5.** If $\phi(x)$ and $\varphi(x)$ satisfies Eq. 12, then for any fixed $P_g$ and $P_r$, $\mathbb{E}_{x \sim P_g}[\phi(f(x))] + \mathbb{E}_{x \sim P_r}[\varphi(f(x))] + \lambda \cdot k(f)^2$ has an lower bound with respect to $f$.

*Proof.*

Given $\exists a, \phi'(a) + \varphi'(a) = 0$, $\phi''(x) \geq 0$ and $\varphi''(x) \geq 0$, we have:

$$
\begin{aligned}
& \mathbb{E}_{x \sim P_g}[\phi(f(x))] + \mathbb{E}_{x \sim P_r}[\varphi(f(x))] + \lambda \cdot k(f)^2 \\
\geq\ & \mathbb{E}_{x \sim P_g}[\phi'(a)(f(x) - a) + \phi(a)] + \mathbb{E}_{x \sim P_r}[\varphi'(a)(f(x) - a) + \varphi(a)] + \lambda \cdot k(f)^2 \\
=\ & \mathbb{E}_{x \sim P_g}[\phi'(a)f(x)] + \mathbb{E}_{x \sim P_r}[\varphi'(a)f(x)] + \lambda \cdot k(f)^2 + c \\
=\ & \phi'(a)[\mathbb{E}_{x \sim P_g}[f(x)] - \mathbb{E}_{x \sim P_r}[f(x)]] + \lambda \cdot k(f)^2 + c \\
\geq\ & \phi'(a)[k \cdot -W_1(P_r, P_g)] + \lambda \cdot k^2 + c \\
=\ & [-\phi'(a)W_1(P_r, P_g)] \cdot k + \lambda \cdot k^2 + c \\
\geq\ & c - \frac{[\phi'(a)W_1(P_r, P_g)]^2}{4\lambda} \qquad\qquad\qquad\qquad\qquad\qquad\qquad \square
\end{aligned}
$$

**Remark.** Theorem 5 implies that there exists an infimum for $\mathbb{E}_{x \sim P_g}[\phi(f(x))] + \mathbb{E}_{x \sim P_r}[\varphi(f(x))] + \lambda \cdot k(f)^2$. According to the definition of infimum, there exists a sequence of $\{f_n\}_{n=1}^{\infty}$ such that $\mathbb{E}_{x \sim P_g}[\phi(f_n(x))] + \mathbb{E}_{x \sim P_r}[\varphi(f_n(x))] + \lambda \cdot k(f_n)^2$ infinitely approaches the infimum.

**Remark.** The Lipschitz constant of $f_n$, i.e., $k(f_n)$, as $n$ goes to infinity, is bounded, because $\mathbb{E}_{x \sim P_g}[\phi(f(x))] + \mathbb{E}_{x \sim P_r}[\varphi(f(x))] + \lambda \cdot k(f)^2 > [-\phi'(a)W_1(P_r, P_g)] \cdot k(f) + \lambda \cdot k(f)^2 + c$.

We further present several simply Lemmas. These Lemmas and their proofs would provide some intuitive impressions on why Eq. 12 is necessary and the properties of proposed objectives.

**Lemma 3.** Assuming $P_g$ and $P_r$ are two delta distributions. If $\phi(x)$ and $\varphi(x)$ satisfies Eq. 12, then for any fixed $P_g$ and $P_r$, $\mathbb{E}_{x \sim P_g}[\phi(f(x))] + \mathbb{E}_{x \sim P_r}[\varphi(f(x))]$ has global minimum with respect to $f$, for any fixed $k(f) = \hat{k}$.

*Proof.* Given $P_g$ and $P_r$ are two delta distributions, according to Theorem 1 and Theorem 2, for $x \sim P_g$ and $y \sim P_r$, $f^*(y) - f^*(x) = \hat{k} \cdot d(x, y)$. Let $f^*(x) = \alpha$ and $\beta = \hat{k} \cdot d(x, y)$, then $f^*(y) = \alpha + \beta$. Define $J_D(\alpha) = \mathbb{E}_{x \sim P_g}[\phi(f(x))] + \mathbb{E}_{x \sim P_r}[\varphi(f(x))] = \phi(\alpha) + \varphi(\alpha + \beta)$.

Given $\exists a$ such that $\phi'(a) + \varphi'(a) = 0$, $\phi''(x) \geq 0$ and $\varphi''(x) \geq 0$, we have, when $\alpha$ is small enough (such that, $\alpha < a$ and $\alpha + \beta < a$), $J_D'(\alpha) = \phi'(\alpha) + \varphi'(\alpha + \beta) \leq \phi'(a) + \varphi'(a) = 0$. Similarly, when $\alpha$ is large enough (such that, $\alpha > a$ and $\alpha + \beta > a$), $J_D'(\alpha) = \phi'(\alpha) + \varphi'(\alpha + \beta) \geq \phi'(a) + \varphi'(a) = 0$.

Therefore, $J_D(\alpha)$ is convex with respect to $\alpha$ and there exists an $\alpha_0$ such that $J_D'(\alpha_0) = 0$, where $J_D$ achieves its the global minimum. When $\phi''(x) > 0$ and $\varphi''(x) > 0$, it is the unique global minimum. $\qquad\square$

**Lemma 4.** Assuming $P_g$ and $P_r$ are two delta distributions. If $\phi(x)$ and $\varphi(x)$ satisfies Eq. 12, then for any fixed $P_g$ and $P_r$, $\mathbb{E}_{x \sim P_g}[\phi(f(x))] + \mathbb{E}_{x \sim P_r}[\varphi(f(x))]$ monotonically increases as $k(f)$ decreases.

*Proof.* Given $P_g$ and $P_r$ are two delta distributions, according to Theorem 1 and Theorem 2, for $x \sim P_g$ and $y \sim P_r$, $f^*(y) - f^*(x) = k \cdot d(x, y)$. Let $f^*(x) = \alpha$ and $\beta = k \cdot d(x, y) > 0$, then $f^*(y) = \alpha + \beta$. Define $J_D(\beta) = \min_{f \in \mathcal{F}_{\text{k-Lip}}} \mathbb{E}_{x \sim P_g}[\phi(f(x))] + \mathbb{E}_{x \sim P_r}[\varphi(f(x))] = \min_\alpha \phi(\alpha) + \varphi(\alpha + \beta)$. We need to prove $J_D(\beta)$ is monotonically decreasing, for $\beta \geq 0$.

Let $0 \leq \beta_1 < \beta_2$, let $\alpha_1 = \min_\alpha \phi(\alpha) + \varphi(\alpha + \beta_1)$ and $\alpha_2 = \min_\alpha \phi(\alpha) + \varphi(\alpha + \beta_2)$. Given $\varphi'(x) < 0$ and $\beta_1 < \beta_2$, we have $\phi(\alpha_1) + \varphi(\alpha_1 + \beta_1) > \phi(\alpha_1) + \varphi(\alpha_1 + \beta_2)$. Given $\alpha_2 = \min_\alpha \phi(\alpha) + \varphi(\alpha + \beta_2)$, we further have $\phi(\alpha_1) + \varphi(\alpha_1 + \beta_1) > \phi(\alpha_1) + \varphi(\alpha_1 + \beta_2) \geq \phi(\alpha_2) + \varphi(\alpha_2 + \beta_2)$. Done.

**Additionally**, with $\phi'(\alpha_1) + \varphi'(\alpha_1 + \beta_1) = 0$ and $\varphi''(x) \geq 0$, we have $\phi'(\alpha_1) + \varphi'(\alpha_1 + \beta_2) \geq 0$. Providing $\phi'(\alpha_2) + \varphi'(\alpha_2 + \beta_2) = 0$, $\phi''(x) \geq 0$ and $\varphi''(x) \geq 0$, we get $\alpha_2 \leq \alpha_1$. That is, $\alpha_2 \leq \alpha_1 < \beta_1 < \beta_2$. When $\phi''(x) > 0$ and $\varphi''(x) > 0$, we have $\alpha_2 < \alpha_1 < \beta_1 < \beta_2$. □

**Lemma 5.** If the support of $P_g$ and $P_r$ is bounded, i.e., $\exists R$ such that $\|x\| < R, \forall x \in \bar{P}_g \cup \bar{P}_r$. Assume $\phi(x)$ and $\varphi(x)$ satisfy Eq. 12 and further have $\phi''(x) > 0$ or $\varphi''(x) > 0$. $\forall N, \exists M$, if $\mathbb{E}_{x \sim P_g}[\phi(f(x))] + \mathbb{E}_{x \sim P_r}[\varphi(f(x))] + \lambda \cdot k(f)^2 < N$, then $|f(x)| < M$, $\forall x \in \bar{P}_g \cup \bar{P}_r$.

*Proof.*

$\exists a, \phi'(a) + \varphi'(a) = 0$ and $\phi''(x) + \varphi''(x) > 0$ implies $\exists b, \phi'(b) + \varphi'(b) > 0$. Then:

$$\mathbb{E}_{x \sim P_g}[\phi(f(x))] + \mathbb{E}_{x \sim P_r}[\varphi(f(x))] + \lambda \cdot k(f)^2$$
$$\geq \mathbb{E}_{x \sim P_g}[\phi'(b)(f(x) - b) + \phi(b)] + \mathbb{E}_{x \sim P_r}[\varphi'(b)(f(x) - b) + \varphi(b)] + \lambda \cdot k(f)^2$$
$$= \mathbb{E}_{x \sim P_g}[\phi'(b)f(x)] + \mathbb{E}_{x \sim P_r}[\varphi'(b)f(x)] + \lambda \cdot k(f)^2 + c$$
$$= [\phi'(b) + \varphi'(b)]\mathbb{E}_{x \sim P_g}[f(x)] + \varphi'(b)[\mathbb{E}_{x \sim P_r}[f(x)] - \mathbb{E}_{x \sim P_g}[f(x)]] + \lambda \cdot k(f)^2 + c$$
$$\geq [\phi'(b) + \varphi'(b)]\mathbb{E}_{x \sim P_g}[f(x)] + \varphi'(b)[W_1(P_r, P_g) \cdot k(f)] + \lambda \cdot k(f)^2 + c$$

$\forall x \in \bar{P}_g \cup \bar{P}_r$, if $f(x) = T$, then: $\forall x \in \bar{P}_g \cup \bar{P}_r$, $f(x) \geq T - k(f) \cdot R$.

$$\mathbb{E}_{x \sim P_g}[\phi(f(x))] + \mathbb{E}_{x \sim P_r}[\varphi(f(x))] + \lambda \cdot k(f)^2$$
$$\geq [\phi'(b) + \varphi'(b)](T - k(f) \cdot R) + \varphi'(b)[W_1(P_r, P_g) \cdot k(f)] + \lambda \cdot k(f)^2 + c$$
$$\geq [\phi'(b) + \varphi'(b)]T - \frac{[\varphi'(b)W_1(P_r, P_g) - [\phi'(b) + \varphi'(b)]R]^2}{4\lambda} + c$$

Given $\mathbb{E}_{x \sim P_g}[\phi(f(x))] + \mathbb{E}_{x \sim P_r}[\varphi(f(x))] + \lambda \cdot k(f)^2 < N$, we have:

$$[\phi'(b) + \varphi'(b)]T - \frac{[\varphi'(b)W_1(P_r, P_g) - [\phi'(b) + \varphi'(b)]R]^2}{4\lambda} + c < N$$
$$\Rightarrow T < (N - c + \frac{[\varphi'(b)W_1(P_r, P_g) - [\phi'(b) + \varphi'(b)]R]^2}{4\lambda}) / [\phi'(b) + \varphi'(b)]$$

Similarly, $\exists a, \phi'(a) + \varphi'(a) = 0$ and $\phi''(x) + \varphi''(x) > 0$ implies $\exists d, \phi'(d) + \varphi'(d) < 0$. And then it implies $T$ is greater than some constant. So, $\exists M$ such that $|f(x)| < M$, $\forall x \in \bar{P}_g \cup \bar{P}_r$. □

# I  PROOF ON THE DUAL FORM OF WASSERSTEIN DISTANCE

We here provide a formal proof for our new dual form of Wasserstein distance. The Wasserstein distance is given as follows:

$$W_1(P_r, P_g) = \inf_{\pi \in \Pi(P_r, P_g)} \mathbb{E}_{(x,y) \sim \pi} [d(x, y)], \tag{19}$$

where $\Pi(P_r, P_g)$ denotes the collection of all probability measures with marginals $P_r$ and $P_g$ on the first and second factors respectively.

The dual form of Wasserstein distance is usually written as:

$$I(P_r, P_g) = \sup_f \mathbb{E}_{x \sim P_r} [f(x)] - \mathbb{E}_{x \sim P_g} [f(x)],$$
$$s.t. \ f(x) - f(y) \leq d(x, y), \ \forall x, \forall y. \tag{20}$$

We will prove that Wasserstein distance in its dual form can also be written as:

$$J(P_r, P_g) = \sup_f \mathbb{E}_{x \sim P_r} [f(x)] - \mathbb{E}_{x \sim P_g} [f(x)],$$
$$s.t. \ f(x) - f(y) \leq d(x, y), \ \forall x \sim P_r, \forall y \sim P_g, \tag{21}$$

which means the constraint in the dual form of Wasserstein distance can be looser than the common formulation.

With the compacted formulation, we argued that a well-defined metric, i.e., $J(P_r, P_g)$, which is equivalent to Wasserstein distance $W_1(P_r, P_g)$ and thus can properly measure the distance between two distributions, may also fail to provide a meaningful $\nabla_x f^*(x)$. This observation indicates that a well-defined distance metric does not necessarily guarantee a meaningful $\nabla_x f^*(x)$ and thus does not guarantee the convergence of $\nabla_x f^*(x)$-based GANs.

## I.1  PROVING THE EQUIVALENCE

**Theorem 6.**  Given $I(P_r, P_g) = W_1(P_r, P_g)$[5], we have $I(P_r, P_g) = J(P_r, P_g) = W_1(P_r, P_g)$

*Proof.*

(i) For any $f$ that satisfies "$f(x) - f(y) \leq d(x, y), \ \forall x, \forall y$", it must satisfy "$f(x) - f(y) \leq d(x, y), \ \forall x \sim P_r, \forall y \sim P_g$". Thus, $I(P_r, P_g) \leq J(P_r, P_g)$.

(ii) Let $F_J = \{f | f(x) - f(y) \leq d(x, y), \ \forall x \sim P_r, \forall y \sim P_g\}$.

Let $A = \{(x, y) | x \sim P_r, y \sim P_g\}$ and $I_A = \begin{cases} 1, (x, y) \in A; \\ 0, otherwise \end{cases}$.

Let $A^c$ denote the complementary set of $A$ and define $I_{A^c}$ accordingly.

$\forall \pi \in \Pi(P_r, P_g)$ We have the following:

$$\begin{aligned} J(P_r, P_g) &= \sup_{f \in F_J} \mathbb{E}_{x \sim P_r} [f(x)] - \mathbb{E}_{x \sim P_g} [f(x)] \\ &= \sup_{f \in F_J} \mathbb{E}_{(x,y) \sim \pi} [f(x) - f(y)] \\ &= \sup_{f \in F_J} \mathbb{E}_{(x,y) \sim \pi} [(f(x) - f(y)) I_A] + \mathbb{E}_{(x,y) \sim \pi} [(f(x) - f(y)) I_{A^c}] \\ &= \sup_{f \in F_J} \mathbb{E}_{(x,y) \sim \pi} [(f(x) - f(y)) I_A] \\ &\leq \mathbb{E}_{(x,y) \sim \pi} [d(x, y) I_A] \\ &\leq \mathbb{E}_{(x,y) \sim \pi} [d(x, y)]. \end{aligned}$$

$J(P_r, P_g) \leq \mathbb{E}_{(x,y) \sim \pi} [d(x, y)], \forall \pi \in \Pi(P_r, P_g)$

$\Rightarrow J(P_r, P_g) \leq \inf_{\pi \in \Pi(P_r, P_g)} \mathbb{E}_{(x,y) \sim \pi} [d(x, y)] = W_1(P_r, P_g)$.

(iii) Combining (i) and (ii), we have $I(P_r, P_g) \leq J(P_r, P_g) \leq W_1(P_r, P_g)$. Given $I(P_r, P_g) = W_1(P_r, P_g)$, we have $I(P_r, P_g) = J(P_r, P_g) = W_1(P_r, P_g)$. □

---

[5]The equivalence of $W_1(P_r, P_g)$ and $I(P_r, P_g)$ is well-known (Villani, 2008; Zemel, 2012).

## I.2    Proving $J(P_r, P_g) = W_1(P_r, P_g)$ Directly

Let $\bar{P}_r$ and $\bar{P}_g$ denote the supports of $P_r$ and $P_g$, respectively. Because $\forall \pi \in \Pi(P_r, P_g)$, it must hold that $\pi(x, y) = 0$ for any $(x, y)$ that outside $\bar{P}_r \times \bar{P}_g$, i.e., $P_r(x) = 0$ or $P_g(y) = 0$ implies $\pi(x, y) = 0$. We let $\Gamma(\bar{P}_r \times \bar{P}_g)$ denote the collection of all probability measures that defined on $\bar{P}_r \times \bar{P}_g$ with marginals $P_r$ and $P_g$ on the first and second factors respectively. Let $M_+(\bar{P}_r \times \bar{P}_g)$ be the collection of all non-negative measures (not necessarily probability measures) on $\bar{P}_r \times \bar{P}_g$.

Before the proof, we would like to give several preliminary notes.

- A more formal and detailed proof for Kantorovich duality **with same logic of justification** can be found in Theorem 2.3 of (Zemel, 2012). And a more relevant version that focused on Wasserstein distance can be found in this blog[6].

- The key change here is that we handle the support of $P_r$ and $P_g$ more carefully, which results in elimination of the unnecessary constraints $f(x) - f(y) \leq d(x, y)$ that involve $(x, y)$ pair where $P_r(x) = 0$ or $P_g(x) = 0$.

- **The validity of the use of the minimax-principle**, i.e., invert the order of $inf$ and $sup$, in this case, is proved in (Zemel, 2012) and the blog.

**Theorem 7.**    $J(P_r, P_g) = W_1(P_r, P_g)$

*Proof.*

$$\inf_{\pi \in \Pi(P_r, P_g)} \mathbb{E}_{(x,y) \sim \pi}[d(x,y)] = \inf_{\pi \in \Gamma(\bar{P}_r \times \bar{P}_g)} \mathbb{E}_{(x,y) \sim \pi}[d(x,y)]$$

$$= \inf_{\pi \in M_+(\bar{P}_r \times \bar{P}_g)} \left[ \int_{\bar{P}_r} \int_{\bar{P}_g} d(x,y)\pi(x,y)dxdy + \begin{cases} 0, & \pi \in \Gamma(\bar{P}_r \times \bar{P}_g) \\ \infty, & otherwise \end{cases} \right]$$

$$= \inf_{\pi \in M_+(\bar{P}_r \times \bar{P}_g)} \left[ \int_{\bar{P}_r} \int_{\bar{P}_g} d(x,y)\pi(x,y)dxdy \right.$$
$$\left. + \sup_f \left[ \mathbb{E}_{s \sim P_r}[f(s)] - \mathbb{E}_{t \sim P_g}[f(t)] - \int_{\bar{P}_r} \int_{\bar{P}_g} (f(x) - f(y))\pi(x,y)dxdy \right] \right]$$

$$= \inf_{\pi \in M_+(\bar{P}_r \times \bar{P}_g)} \sup_f \left[ \mathbb{E}_{s \sim P_r}[f(s)] - \mathbb{E}_{t \sim P_g}[f(t)] + \int_{\bar{P}_r} \int_{\bar{P}_g} [d(x,y) - (f(x) - f(y))]\pi(x,y)dxdy \right]$$

$$= \sup_f \inf_{\pi \in M_+(\bar{P}_r \times \bar{P}_g)} \left[ \mathbb{E}_{s \sim P_r}[f(s)] - \mathbb{E}_{t \sim P_g}[f(t)] + \int_{\bar{P}_r} \int_{\bar{P}_g} [d(x,y) - (f(x) - f(y))]\pi(x,y)dxdy \right]$$

$$= \sup_f \left[ \mathbb{E}_{s \sim P_r}[f(s)] - \mathbb{E}_{t \sim P_g}[f(t)] + \inf_{\pi \in M_+(\bar{P}_r \times \bar{P}_g)} \int_{\bar{P}_r} \int_{\bar{P}_g} [d(x,y) - (f(x) - f(y))]\pi(x,y)dxdy \right]$$

$$= \sup_f \left[ \mathbb{E}_{s \sim P_r}[f(s)] - \mathbb{E}_{t \sim P_g}[f(t)] + \begin{cases} 0, & f(x) - f(y) \leq d(x,y), \ \forall x \sim P_r, \forall y \sim P_g \\ -\infty, & otherwise \end{cases} \right]$$

$$= \sup_f \mathbb{E}_{s \sim P_r}[f(s)] - \mathbb{E}_{t \sim P_g}[f(t)], \ s.t. \ f(x) - f(y) \leq d(x,y), \ \forall x \sim P_r, \forall y \sim P_g. \qquad \square$$

## I.3    Another perspective: eliminating redundant constraint

To provide a more comprehensive understanding on why constraint $f(x) - f(y) \leq d(x, y)$ that involves point $(x, y)$ where $P_r(x) = 0$ or $P_g(x) = 0$ is unnecessary, we here give a intuitive explanation in the following way: one can safely remove all constraints that does not involve any point in the support of $P_g$ and $P_r$, because give $f(x) - f(y) \leq d(x, y), \forall x \sim P_r, \forall y \sim P_g$, it is sufficient to bound the value of $f(x)$ for $x \sim P_r$ and $f(y)$ for $y \sim P_g$. In other words, given the existence of constraint $f(x) - f(y) \leq d(x, y), \forall x \sim P_r, \forall y \sim P_g$, the other constraints can be safely eliminated without affecting the final solution.

---

[6] https://vincentherrmann.github.io/blog/wasserstein/

## J    CONNECTIONS WITH OPTIMAL TRANSPORT

The $1^{st}$-Wasserstein distance, also named as the Earth Mover's distance, is a special form of optimal transport (Villani, 2008), which measures the minimal cost of moving the source distribution to the target distribution, and the optimal coupling $\pi(x, y)$ describes the transport plan, *i.e.* how much density we should move from $x$ to $y$. Naturally, updating the generator according to the optimal coupling would pull $P_g$ towards $P_r$.

However, updating the generator according to the optimal coupling $\pi(x, y)$ is a totally different mechanism for training GANs. In typical GANs, we update the generator following $\nabla_x f^*(x)$. An interesting fact[7] is that: with Wasserstein GAN objective, when updating the generator according to $\nabla_x f^*(x)$, it follows the optimal coupling $\pi$, if (and only if) there is Lipschitz constraint and the $d(x, y)$ represents the Euclidean distance.

In general optimal transport, $d(x, y)$ is not required to be a distance and can be any cost function. To the best knowledge of the authors, it is hard to access the coupling information from $\nabla_x f^*(x)$ if $d(x, y)$ is arbitrary. However, fortunately, given the optimal coupling $\pi$, directly updating each sample towards its target is also possible. An instance of this line of work can be found in (Sanjabi et al., 2018), where the objective (Seguy et al., 2017) of generator is $\mathbb{E}_{x \sim P_g} \left[ \mathbb{E}_{y \sim \pi(\cdot|x)} \left[ d(x, y) \right] \right]$.

In summary, we think training GANs with optimal mapping and with Lipschitz constraint are two mechanisms with different underlying principles, and Wasserstein GAN in the Lipschitz dual form with Euclidean distance is the connecting point.

## K    HYPER-PARAMETER & NETWORK ARCHITECTURE

We follow the network architecture proposed in (Gulrajani et al., 2017) to conduct our experiments on CIFAR-10, Tiny Imagenet, Oxford 102. The details of network architecture are in Table 3.

Generator:

| Operation | Kernel | Resample | Output Dims |
|---|---|---|---|
| Noise | N/A | N/A | 128 |
| Linear | N/A | N/A | 128×4×4 |
| Residual block | 3×3 | UP | 128×8×8 |
| Residual block | 3×3 | UP | 128×16×16 |
| Residual block | 3×3 | UP | 128×32×32 |
| Conv & Tanh | 3×3 | N/A | 3×32×32 |

Critic:

| Operation | Kernel | Resample | Output Dims |
|---|---|---|---|
| Residual Block | 3×3×2 | Down | 128×16×16 |
| Residual Block | 3×3×2 | Down | 128×8×8 |
| Residual Block | 3×3×2 | N/A | 128×8×8 |
| Residual Block | 3×3×2 | N/A | 128×8×8 |
| ReLU,mean pool | N/A | N/A | 128 |
| Linear | N/A | N/A | 1 |

| Optimizer: Adam with beta1=0.0, beta2=0.9; |
|---|
| For more details, please refer to our published codes. |

Table 3: Hyper-parameter and Network Architectures

---

[7]Assuming $f^*(x)$ is differentiable.

