# OpenReview forum: "Understanding the Effectiveness of Lipschitz-Continuity in Generative Adversarial Nets"
_ICLR.cc/2019/Conference_

### Official Review · AnonReviewer3 · 2018-11-01
**Although the proposed general formulation is itself interesting, some of the arguments are not sound, and the proposed scheme is somehow similar to the gradient-penalty-based formulation in Gulrajani et al. (2017).**

**Rating:** 5
**Confidence:** 4

**Review:**


[pros]
- It proposes a general formulation of GAN-type adversarial learning as in (1), which includes the original GAN, WGAN, and IPM-type metrics as special cases.
- It also proposes use of the penalty term in terms of the Lipschitz constant  of the discriminative function.

[cons]
- Some of the arguments on the Wasserstein distance and on WGAN are not sound.
- Theorem 3 does not make sense.
- The proposed scheme is eventually similar to the gradient-penalty-based formulation in Gulrajani et al. (2017).

[Quality]
I found some weaknesses in this paper, so that I judge the quality of this paper not to be high. For example, the criticisms on the Wasserstein distance in Section 2.3 and in Section 4.4, as well as the argument on WGAN at the end of Section 3.1, is not sound. The claim in Theorem 3 does not make sense, if we literally take its statement. All these points are detailed below.

[Clarity]
The main paper is clearly written, whereas in the appendices I noticed several grammatical and spelling errors as well as unclear descriptions.

[Originality]
Despite that the arguments in this paper are interesting, the proposed scheme is somehow eventually similar to the gradient-penalty-based formulation in Gulrajani et al. (2017), with differences being introduction of loss metrics $\phi,\varphi,\psi$ and the form of the gradient penalty, $\max \|\nabla f(x)\|_2^2$ in this paper versus $E[(\|\nabla f(x)\|_2-1)^2]$ in Gulrajani et al. (2017). This fact has made me to think that the originality of this paper is marginal.

[Significance]
This paper is significant in that it would stimulate empirical studies on what objective functions and what types of gradient penalty are efficient in GAN-type adversarial learning.

Detailed comments:

In Section 2.3, the authors criticize use of the Wasserstein distance as the distance function of GANs, but their criticism is off the point. It is indeed a problem not of the Wasserstein distance itself, but of its dual formulation.

It is true mathematically that $f$ in equation (8) does not have to be defined outside the supports of $P_g$ and $P_r$ because it does not affect the expectations in (8). In practice, however, one may regard that $f$ satisfies the condition $f(x)-f(y)\le d(x,y)$ not only on the supports of $P_g$ and $P_r$ but throughout the entire space $\mathbb{R}^n$. It is equivalent to requiring $f$ to satisfy the 1-Lipschitz condition on $\mathbb{R}^n$, and is what WGAN (Arjovsky et al., 2017) tries to do in its implementation of the "critic" $f$ via a multilayer neural network with weight clipping.

One can also argue that, if one defines $f$ only on the supports of $P_g$ and $P_r$, then it should trivially be impossible to obtain gradient information which can change the support of $P_g$. The common practice of requiring the Lipschitz condition throughout $\mathbb{R}^n$ is thus reasonable from this viewpoint. This is therefore not the problem of the Wasserstein distance itself, but the problem regarding how the dual problem is implemented in learning of GANs. In this regard, the discussion in this section, as well as that in Section 4.4, is misleading.

On optimizing $k$, I do not agree with the authors's claim at the end of Section 3.1 that WGAN may not have zero gradient with respect to $f$ even when $P_g=P_r$. Indeed, when $P_g=P_r$, for any measurable function $f$ one trivially has $J_D[f]=E_{x\sim P_g}[f(x)]-E_{x\sim P_r}[f(x)]=0$, so that the functional derivative of $J_D$ with respect to $f$ does vanish identically.

I do not understand the claim of Theorem 3. I think that the assumption is too strong. If one literally takes "$\forall x \not= y$", then one can exchange $x$ and $y$ in the condition $f(y)-f(x)=k\|x-y\|$ to obtain $f(x)-f(y)=k\|y-x\|$, which together would imply $k=0$, and consequently $f$ is constant. One would be able to prove that if there exists $(x,y)$ with $x \not= y$ such that $f(y)-f(x)=k\|x-y\|$ holds then the gradient of $f$ at $x_t$ is equal to $k(y-x)/\|x-y\|$ under the Lipschitz condition.

Appendix G: Some notations should be made more precise. For example, in the definition of J_D the variable of integration $x$ has been integrated out, so that $J_D$ no longer has $x$ as its variable. The expression $\partial J_D/\partial x$ does not make any sense. Also, $J_D^*(k)$ is defined as "arg min" of $J_D$, implying as if $J_D^*(k)$ were a $k$-Lipschitz function.

Page 5, line 36: $J_D(x)$ appears without explicit definition.

Page 23, lines 34 and 38: Cluttered expression $\frac{\partial [}{\partial 2}]$ makes the statements not understandable. It also appears on page 24 several times.

---

> ### Author Response · Authors · 2018-11-16
> **Response to Reviewer 3**
>
> Thanks for your constructive feedback.
>
> Q: In Section 2.3, the authors criticize the use of the Wasserstein distance as the distance function of GANs, but their criticism is off the point. This is not the problem of the Wasserstein distance itself, but the problem regarding how the dual problem is implemented in learning of GANs. In practice, however, one may regard that $f$ satisfies the condition $f(x)-f(y)\le d(x,y)$ not only on the supports of $P_g$ and $P_r$ but throughout the entire space $\mathbb{R}^n$.
>
> >> We have refined our statements to make it more rigorous. We argued that Wasserstein distance in the dual form with compacted constraint will fail to provide valid gradients. We have revised related sections. It is much clearer now.
>
> >> The practical solution of WGAN, which usually involves Lipschitz constraint, is sound. What we want to emphasize is that the explanation on the underlining working mechanism in the Wasserstein GAN paper does not hold very well. It is not Wasserstein distance which can properly measure the distance that makes it work, but the Lipschitz constraint. As we showed in the paper, in the compact dual form (without Lipschitz) of Wasserstein distance, $\nabla_x f*(x)$ may also fail to provide a meaningful gradient; and with Lipschitz, lots of GAN objectives can guarantee a meaningful gradient in terms of $\nabla_x f*(x)$, not limited to Wasserstein distance.
>
> Q: The claim in Theorem 3 does not make sense. If we literally take its statement, it would imply $k=0$, and consequently, $f$ is constant. One would be able to prove that if there exists $(x,y)$ with $x \not= y$ such that $f(y)-f(x)=k\|x-y\|$ holds then the gradient of $f$ at $x_t$ is equal to $k(y-x)/\|x-y\|$ under the Lipschitz condition.
>
> >> Sorry, this is a typo. We indeed mean “$\forall x \not= y$, if $f(y)-f(x)=k\|x-y\|$” as what you have guessed, but it was miswritten as “if $f(y)-f(x)=k\|x-y\|$, $\forall x \neq y$”. We have revised it in our new version.
>
> Q: On optimizing $k$, I do not agree with the authors' claim at the end of Section 3.1 that WGAN may not have zero gradient with respect to $f$ even when $P_g=P_r$. Indeed, when $P_g=P_r$, for any measurable function $f$ one trivially has $J_D[f]=E_{x\sim P_g}[f(x)]-E_{x\sim P_r}[f(x)]=0$, so that the functional derivative of $J_D$ with respect to $f$ does vanish identically.
>
> >> Sorry for the typo. It should be ‘‘$\nabla_x f*(x)$ may not be zero’’. We have miswritten it as $\nabla_f*$. For WGAN, when $P_g=P_r$, f* can be arbitrary, so $\nabla_x f*(x)$ may not be zero. We have revised it in our new version.
>
> ----- Minor Points -----
>
> Q: Page 5, line 36: $J_D(x)$ appears without explicit definition.
>
> >> Thanks for pointing this out. We have added the definition for $J_D(x)$ in the revised version.
>
> Q: Appendix G: Some notations should be made more precise. For example, in the definition of $J_D$ the variable of integration $x$ has been integrated out so that $J_D$ no longer has $x$ as its variable. The expression $\partial J_D/\partial x$ does not make any sense.
>
> >> By $\partial J_D/\partial x$, we want to refer to the term that is being integrated over $x$. In the current form, it is analogous to the gradient of indefinite integral. i.e., F = \int_x f(x) dx and F’ = f(x). We have replaced the confusing derivative form with a simple notation.
>
> Q: Appendix G: $J_D^*(k)$ is defined as "arg min" of $J_D$, implying as if $J_D^*(k)$ were a $k$-Lipschitz function.
>
> >> $J_D^*(k)$ should be defined as the min value, not arg min. We have corrected it.
>
> Q: Page 23, lines 34 and 38: cluttered expression $\frac{\partial [}{\partial 2}]$ makes the statements not understandable. It also appears on page 24 several times.
>
> >> The $\frac{\partial [}{\partial 2}]$ comes from a breaking \newcommand for second-order derivation in latex. We have fixed it.
>
> Thanks a lot for the careful reading of our paper and the detailed comments, which are very helpful.

---

### Official Review · AnonReviewer2 · 2018-11-02
**Review for "Understanding the Effectiveness of Lipschitz-Continuity in Generative Adversarial Nets"**

**Rating:** 4
**Confidence:** 4

**Review:**

The authors try to claim that Lipschitz continuity of the discriminator is a fundamental solution of GANs, and that current methods do not satisfy this approach in principle.

There are several false statements in this paper. In particular, sections 2.3 and 4.4 are wrong (and most of the paper is based on statements made there). The necessary constraint for the Wasserstein distance is NOT f(x) - f(y) <= d(x, y) for all x ~ Pr, y ~ Pg. It has to actually be 1-Lipschitz in the entire space. See Chapters 5 and 6 of [1], for example remark 6.4 or particular cases 5.16 and 5.4. Indeed, this is how it is written in all of the literature this reviewer is aware off, and it's a fact well used in the literature. Indeed, all the smoothness results for optimal transport in [1] heavily exploit the fact that the gradient of the critic is in the direction of the optimal transport map, which wouldn't be the case in the situation the authors try to claim of 'f not being defined outside of the support of Pr or Pg'.

Furthermore, the relationship between Lipschitz continuity and having a gradient is elaborated in [2] https://arxiv.org/abs/1701.07875 , for example figure 2 clearly show this. Furthermore, and contrary to what section 4.5 tries to claim, the idea that most conclusions of wgan hold *without* the Wasserstein distance, but with Lipschitz continuity are already elaborated in the wgan paper. See in fact, appendix G.1 [3], where this is described in detail.

[1]: http://cedricvillani.org/wp-content/uploads/2012/08/preprint-1.pdf
[2]: http://proceedings.mlr.press/v70/arjovsky17a/arjovsky17a.pdf
[3]: http://proceedings.mlr.press/v70/arjovsky17a/arjovsky17a-supp.pdf

---

> ### Author Response · Authors · 2018-11-16
> **Response to Reviewer 2 (2/2)**
>
> Thanks for your constructive feedback.
>
> Q: The idea that most conclusions of WGAN hold *without* the Wasserstein distance, but with Lipschitz continuity are already elaborated in the WGAN paper. See in fact, Appendix G.1 [3], where this is described in detail.
>
> >> Given Wasserstein GAN is one of the most important papers in GAN community and we have realized that its argument in the main text does not hold very well, we believe it is necessary to highlight the point. Currently, most people tend to believe that a good distance metric is the key to the convergence or stability of GANs. One contribution of our paper is that it thoroughly expounded that: the property of the gradient in terms of \nabla_x f*(x) is substantially different from the property of a distance metric; and to ensure the convergence of GANs or design new formulation for GANs, one should carefully check whether the gradient \nabla_x f*(x) is reliable.
>
> >> Regarding Appendix G.1 [3], although it states that Lipschitz might be generally applicable, the discussion there is far from enough: (i) the discussion in Appendix G.1 is limited to the objective of the original GAN; our theorem, in contrast, elaborated a family of GAN objectives and characterized the necessary condition where Lipschitz condition ensures the convergence. (ii) the discussion in Appendix G.1 ignores the $\log$ term in the objective of original GAN; our theorem is, however, directly applicable to the whole objective of the original GAN.* (iii) according to our theorem, for any objective other than W-distance, it is theoretically necessary to penalize the Lipschitz constant k(f) to ensure the convergence; though Appendix G.1 mentioned that to avoid the saturation, k need to be small, it fails to cover the other fold that ‘‘\nabla_x f*(x)=0 for all x’’ might also happen even if there is no saturation region or saturation region is not touched.**
>
> [1]: Optimal transport, old and new
> [2]: http://proceedings.mlr.press/v70/arjovsky17a/arjovsky17a.pdf
> [3]: http://proceedings.mlr.press/v70/arjovsky17a/arjovsky17a-supp.pdf
> [4]: Improved Training of Wasserstein GANs
>
> * In Appendix G.1, it discusses the properties of f with bounded value-range (to simulate a classifier), while in our paper, f is assumed to have unbounded value range and loss metrics are applied to the unbounded f. Therefore, the arguments are actually quite different.
>
> ** That is to say, small k is not enough to guarantee the convergence, and penalizing/decreasing the Lipschitz constant is necessary. Given a fixed Lipschitz constant k, according to our analysis, the following state is possible: ‘‘Pg!=Pr’’ and  ‘‘for each x, f*(x) is optimal’’, but there does not exist two points x,y such that |f(x)-f(y)|>k|x-y|. In this case, ‘‘\nabla_x f*(x)=0 for all x’’ and the generator stop learning, however, Pg does not equal to Pr.

---

> ### Author Response · Authors · 2018-11-16
> **Response to Reviewer 2 (1/2)**
>
> Thanks for your constructive feedback.
>
> Q: There are several false statements in this paper. In particular, sections 2.3 and 4.4 are wrong. The necessary constraint for the Wasserstein distance is NOT f(x) - f(y) <= d(x, y) for all x ~ Pr, y ~ Pg. It has to actually be 1-Lipschitz in the entire space. Indeed, this is how it is written in all of the literature this reviewer is aware off, and it's a fact well used in the literature.
>
> >> We are sorry for not providing a formal proof of this in the paper. Please see Appendix I.1 in the newly uploaded version of our paper where we strictly prove that the new dual form of Wasserstein distance is valid.
>
> >> We have carefully checked a few related materials and their proofs on the Kantorovich duality. And we realize that they basically assume the supports of the two distribution are the entire space (or they just ignore the things about support). But the fact is that once you handle the supports of distributions delicately, you will get a looser constraint. See Appendix I.2 for details.
>
> >> We would also like to provide an intuitive explanation on why “f(x) - f(y) <= d(x, y) for all x ~ Pr, y ~ Pg” is enough. Taking “Pr and Pg are both delta distributions with the support being a simple point x0 and y0, respectively” as the instance would make it much clear. Given f(x0)-f(y0)<=|x0-y0| is enough to get the Wasserstein distance |x0-y0|. Generally, “f(x) - f(y) <= d(x, y) for all x ~ Pr, y ~ Pg”, comparing with 1-Lipschitz in the entire space, only eliminates these “f(x) - f(y) <= d(x, y)” that involve x not in Pr or y not in Pg. These constraints are redundant and do not affect the final solution.
>
> >> We have rewritten section 2.3, section 4.4 and all related parts in the paper. Now it should have more strictly stated our observation, which will be less confusing.
>
> Q: All the smoothness results for optimal transport in [1] heavily exploit the fact that the gradient of the critic is in the direction of the optimal transport map, which wouldn't be the case in the situation the authors try to claim of 'f not being defined outside of the support of Pr or Pg'.
>
> >> Yeah. We knew that theorem. Recent GAN paper [4] also highlighted this property in its Proposition 1. And we have also mentioned this theorem in our paper in Section 3.1 stated in Eq. 11.
>
> >> The thing is that: these theorems all use the 1-Lipschitz as the constraint, therefore, f* and its gradient is well-defined. We actually argued that there exists a dual form of Wasserstein distance where the constraint is looser than 1-Lipschitz; and when the compacted constraint is applied (not 1-Lipschitz), f* is not defined outside the supports of Pg and Pr, and its gradient is also likely to be ill-behaving. We have made these arguments more strict in the revised version.
>
> Q: Furthermore, the relationship between Lipschitz continuity and having a gradient is elaborated in [2].
>
> >> Our theorem is a more general version of the theorem you mentioned, where we no longer restrict the objective to be Wasserstein distance and state the general properties of optimal discriminative function f* under Lipschitz-continuity condition.

---

> > ### Comment · AnonReviewer2 · 2018-11-27
> > **Response to rebuttal**
> >
> > "And we realize that they basically assume the supports of the two distribution are the entire space (or they just ignore the things about support)."
> > This is false! They assume the duality functions are defined in the entire space, but not necessarily the distributions! This is why figure 1, b) doesn't make any sense, since the Kantorovich dual is defined in the *entire* space.
> >
> > "We would also like to provide an intuitive explanation on why “f(x) - f(y) <= d(x, y) for all x ~ Pr, y ~ Pg” is enough. " This is obvious, all you are doing is rewriting the Wasserstein distance in a way that the gradient of W(Pr, Ptheta) is not the average gradient of the critic on the generator's samples. This is not a flaw of the Wasserstein distance, since *the gradient of the Wasserstein distance wrt the generator's parameters itself is well defined and well behaved*, but the gradient with respect to your particular dual formulation is not. If the authors want to make the claim that not all dual formulations satisfy an envelope theorem (as in this pathological formulation, since the space of critic {|f(x) - f(y)| <= d(x,y) for all x ~ Pr, y ~ Ptheta} changes with theta, which is also contrary to what we call a dual formulation typically), this is the claim the authors should be making, not that the Wasserstein distance itself is broken. The authors try to claim in multiple moments sentences like "the gradient of f∗(x) from the dual form of Wasserstein distance given a compacted
> > dual constraint also does not reflect any useful information about other points in P". This is not *the* dual formulation that the entire community uses (aka the Kantorovich duality), but a particular dual-like formulation that the author's concocted which breaks.
> >
> > I think the changes made to the paper are a good direction in rewriting the claims (hence my changed score), but this is still far for sufficient in terms of contribution and technical correctness for a conference like ICLR.

---

> > > ### Author Response · Authors · 2018-11-27
> > > **We truly appreciate your feedback. Let’s continue the discussion.  (2/2)**
> > >
> > >
> > > Q: The space of critic {|f(x) - f(y)| <= d(x,y) for all x ~ Pr, y ~ Ptheta} changes with theta, which is also contrary to what we call a dual formulation typically.
> > >
> > > >> ‘‘which is also contrary to what we call a dual formulation typically’’: can you provide any reference to support this argument? Furthermore, no matter what you call our new dual form, it does not change the fact that it is equivalent to the Wasserstein distance in primal form and can properly measure the distance, and the fact that it does not guarantee the convergence, which can well support our argument that ‘‘a well-defined distance metric does not necessarily guarantee the convergence of GANs’’.
> > >
> > > Q: The authors try to claim in multiple moments sentences like "the gradient of f∗(x) from the dual form of Wasserstein distance given a compacted dual constraint also does not reflect any useful information about other points in P". This is not *the* dual formulation that the entire community uses (aka the Kantorovich duality), but a particular dual-like formulation that the author's concocted which breaks.
> > >
> > > >> Exactly, but this argument is off the point. Our argument is based on the compacted dual form, not the formulation that the entire community uses. What is the problem here? We are not criticizing the typical formulation that the entire community uses. We are criticizing that a well-defined distance metric does not necessarily guarantee the convergence of GANs, and particularly, Wasserstein distance in a new dual form also does not guarantee the convergence.
> > >
> > > >> ‘‘author's concocted which breaks’’: why do you claim it breaks, since we have proved their equivalence in Theorem 6? We think we need to be rigorous in the rebuttal. Could you specify your reason?
> > >
> > > Q: "And we realize that they basically assume the supports of the two distribution are the entire space (or they just ignore the things about support)." This is false! They assume the duality functions are defined in the entire space, but not necessarily the distributions!
> > >
> > > >> Thanks for your points. Given that we have agreed that the new dual form is valid, and this is only relevant to a minor support of ‘‘why the new dual form is valid’’ and is not included in the paper (only appears in the rebuttal comments and can be safely deleted), let’s discuss this later and address the problem about Wasserstein distance first. **It does not mean we agree with the points. It is not an important point that deserves further clarification/argument at this very moment. Let’s just ignore it for now.**
> > >
> > > Q: I think the changes made to the paper are a good direction in rewriting the claims (hence my changed score), but this is still far for sufficient in terms of contribution and technical correctness for a conference like ICLR.
> > >
> > > >> Thanks for the positive comments and changing the score, but we are wondering what the technical correctness issue is in our paper. Could you specify it? Then we can have further discussion and clarification.
> > >
> > > Thanks again for your valuable comments. Let’s continue the discussion.

---

> > > > ### Comment · AnonReviewer2 · 2018-11-27
> > > > **Clarification**
> > > >
> > > > Perfect, let me be as formal as I can be. There are several quantities in here involved. Let F = {f : X -> R, 1-Lip in X}. Let H_theta = {f: X -> R, |f(x) - f(y) | <= d(x, y) with x ~Pr, y ~Ptheta with probability 1}. Let L(f, theta) = E_{x ~ Pr} [f(x)] - E_{y ~ Pz} E[f(g_theta(z))].
> > > >
> > > > I) W(Pr, Ptheta)
> > > > II) Max_{f in F} L(f, theta)
> > > > III) Max_{f in H_theta} L(f, theta)
> > > >
> > > > Facts:
> > > > - (I) = (II), this is the Kantorovich duality.
> > > > - (II) = (III), this you proved.
> > > > - Following grad W(Pr, Ptheta) guarantees convergence as much as one can guarantee convergence of local minimization for locally Lipschitz functions. This is, with an arbitrarily small mollifier one can give convergence to a first order saddle point. This is because W(Pr, Ptheta) is itself locally Lipschitz.
> > > > - If f* is the minimizer in (II), then E_z[grad_theta f*(g_theta(z))] = grad W(Pr, Ptheta) by an envelope theorem as shown in WGAN, thus following this guarantees convergence in the same way as one can do it for W(Pr, Ptheta) itself.
> > > > - If h* is the minimizer in (III), then following E_z[grad_theta h*(g_theta(z))] does not guarantee convergence.
> > > >
> > > > Ergo, what you showed is that approximating (III) is a consistent estimator of the Wasserstein distance, and that gradients of the estimator do not give a consistent estimator of the gradient of the Wasserstein distance. This is not much of a meaningful contribution, since it's widely known that the gradient of an estimator is not necessarily an estimator of the gradient unless one imposes constraints such as bounded Sobolev norms. The same problem appears in e.g. actor critic formulations or other problems like synthetic gradients (see Sobolev training of neural networks for instance). The formulation in (III) was introduced by you, and thus showing the failure of the gradient of it as a gradient estimator is pretty meaningless unless it highlights problems of the formulations that other people use in practice.
> > > >
> > > > Thus, the problems you showcase are not problems of the Wasserstein distance itself, are problems of the estimator you presented of the Wasserstein distance. One can do this with any quantity that can be rephrased as a maximization problem: estimating it and taking gradients is not the same as estimating its gradient. This is not a problem of the quantity itself but of the estimator. If the estimator was widely used this would be a meaningful contribution, but as you show the estimator being used in practice does not actually suffer from the problems of the estimator you introduced.

---

> > > > > ### Author Response · Authors · 2018-11-28
> > > > > **Thank you. Nice progress. (2/2)**
> > > > >
> > > > >
> > > > > 2. What is trivial and what is not?
> > > > >
> > > > > >> The trivial thing is that: if two functions are identical, then their gradient (with respect to the input) are equal. That is, the gradient of a perfect estimator of an objective is equal to the gradient of the objective.
> > > > >
> > > > > >> The non-trivial (not noticed by many people) thing is that: the gradient of a helper of the estimator of an objective (e.g., the optimal discriminative function f* of Wasserstein distance in dual form) is not equal to the gradient of the objective. This is what we argued in the paper, i.e., the gradient of the optimal discriminative function mostly has nothing to do with the gradient of the objective. Note that the optimal discriminative function is **only a helper** of estimator and is not equivalent to the estimator itself.
> > > > >
> > > > > 3. Specific response to your comments:
> > > > >
> > > > > Q: It's widely known that the gradient of an estimator is not necessarily an estimator of the gradient unless one imposes constraints such as bounded Sobolev norms. The same problem appears in e.g. actor critic formulations or other problems like synthetic gradients (see Sobolev training of neural networks for instance).
> > > > >
> > > > > >> We have checked that paper ‘‘Sobolev training of neural networks’’. Could you specify which sentence or theorem in that paper provides any support for your claim? It is simply making use of the trivial fact that we highlighted in ‘‘2. What is trivial and what is not?’’ to improve the training. It definitely makes sense and has nothing to do with our arguments here.
> > > > >
> > > > > Q: One can do this with any quantity that can be rephrased as a maximization problem: estimating it and taking gradients is not the same as estimating its gradient. This is not a problem of the quantity itself but of the estimator.
> > > > >
> > > > > >> The first sentence is correct. This is what we have argued. The second sentence is, however, incorrect. It is not a problem about the estimator or the quantity. As long as the two are equal, their gradient are equal.
> > > > >
> > > > > >> It is about what you are taking the derivative of. That is, the problem lies in the subject that you are taking the derivative of is not the estimator but the optimal function which is only a building block of the estimator. If you can somehow take derivative of the estimator, the gradient is definitely right. The problem in GANs is that, usually, you are not able to take the derivative of the objective that you are trying to estimate.
> > > > >
> > > > > Q: The formulation in (III) was introduced by you, and thus showing the failure of the gradient of it as a gradient estimator is pretty meaningless unless it highlights problems of the formulations that other people use in practice.
> > > > >
> > > > > >> Clarification: we are not criticizing the typical formulation that the community uses. We are criticizing that a well-defined distance metric does not necessarily guarantee the convergence of GANs, and particularly, Wasserstein distance in a new dual form also does not guarantee the convergence.
> > > > >
> > > > > >> If you wish, you could say ‘‘criticizing a well-defined distance metric does not necessarily guarantee the convergence of GANs is pretty meaningless’’.

---

> > > > > ### Author Response · Authors · 2018-11-28
> > > > > **Thank you. Nice progress. (1/2)**
> > > > >
> > > > > We sincerely thank you for your prompt reply.
> > > > >
> > > > > In your clarification, you mainly try to claim that ‘‘showing the failure of a new dual form of Wasserstein distance is meaningless’’, ‘‘not much of a meaningful contribution’’,  ‘‘ it's widely known’’, etc. So, are we now talking about the contribution? Nice progress.
> > > > >
> > > > > Above all, let’s make sure one thing: do you still think there is any technical correctness issue in our paper? If so, please specify it. Otherwise, we would take it as we have convinced you that there is no technical correctness issue in our paper.
> > > > >
> > > > > For contribution clarification, we have listed the contributions of this paper in the uppermost comment. Please check it out and we hope that you might notice that the specific contribution that you does not recognize is only a single item in the long list. Nonetheless, we will respond to your clarification and address this specific concern in the next.
> > > > >
> > > > > We have to point out that there is a serious mistake in your statements. Let’s detail it as follows:
> > > > >
> > > > > 1. Theorem 3 in Wasserstein GAN [1] is a false Theorem. And the 4th fact you provided in the clarification does not hold.
> > > > >
> > > > > >> The proof in [1] essentially uses the Envelope theorem [2] to argue that the gradient of Wasserstein distance in primal is equal to the gradient of Wasserstein distance in dual form. However, in the proof, it ignores the fact that the optimal discriminative function is actually a function of the parameter of Pg and ignores the relevant gradient.* In Envelope theorem, it is critical to writing the objective as a simple function of the parameter, where all other variables need to be written as functions of the parameter and the gradients of all terms are unignorable. Please check the first Theorem in [2].
> > > > >
> > > > > >> From another aspect (proof by contradiction): if the argument in the proof of Theorem 3 in Wasserstein GAN is correct, one can easily extend it to any dual form of Wasserstein distance, which implies ‘‘the gradient of W(Pr, Pg) is also equal to the gradient of our new dual form’’. It is contradictory to our analysis. Please re-check Section 2.3, where the argument is straightforward and is definitely true.
> > > > >
> > > > > >> As it stands, the correct argument about the validity of the gradient of Wasserstein distance in dual form with Lipschitz-continuity condition is Proposition 1 in [3] or our theorems.
> > > > >
> > > > > [1] https://arxiv.org/pdf/1701.07875.pdf
> > > > > [2] https://en.wikipedia.org/wiki/Envelope_theorem
> > > > > [3] Improved Training of Wasserstein GANs
> > > > >
> > > > > * This turns out to be the same type of mistake that Martin Arjovsky et al. have made in Theorem 2.5 of their paper ‘‘Towards Principled Methods for Training Generative Adversarial Networks’’ (ICLR 2017). This is just for your information, not important here. If you are interested, we can have more discussions on this after the rebuttal.

---

> > > > > > ### Comment · AnonReviewer2 · 2018-11-28
> > > > > > **Reply**
> > > > > >
> > > > > > "1. Theorem 3 in Wasserstein GAN [1] is a false Theorem. And the 4th fact you provided in the clarification does not hold. "
> > > > > >
> > > > > > This is ridiculous.
> > > > > >
> > > > > > "The proof in [1] essentially uses the Envelope theorem [2] to argue that the gradient of Wasserstein distance in primal is equal to the gradient of Wasserstein distance in dual form. However, in the proof, it ignores the fact that the optimal discriminative function is actually a function of the parameter of Pg and ignores the relevant gradient.*"
> > > > > >
> > > > > > It does not! The envelope theorem is meant EXACTLY for those situations. See the same wikipedia article, or the proof of theorem 3 of wgan in detail. In the notation of the wikipedia article, X = {1-Lip functions}, and x \in X* would be the optimal critic. It is obvious that this changes with the discriminator, but the derivative of the loss wrt the critic is 0 when the critic is optimal so the corresponding term to the changes of the critic vanishes. I have no interest in discussing this further, the authors are not paying enough attention or rigour to their arguments.
> > > > > >
> > > > > > "From another aspect (proof by contradiction): if the argument in the proof of Theorem 3 in Wasserstein GAN is correct, one can easily extend it to any dual form of Wasserstein distance, which implies ‘‘the gradient of W(Pr, Pg) is also equal to the gradient of our new dual form’’. It is contradictory to our analysis. Please re-check Section 2.3, where the argument is straightforward and is definitely true. "
> > > > > >
> > > > > > NO, one can't if the dual space itself X varies on theta (such as with {|f(x) - f(y)| <= d(x, y) for x ~ Pr, y ~ Ptheta with prob 1}, and not with the space of 1-Lipschitz functions), such as in your case.
> > > > > >
> > > > > > =================================================
> > > > > >
> > > > > > As a last comment, if the area chairs think my arguments are wrong, feel free to disregard what I said. If I am wrong, then sadly figuring it out from the paper and this discussion is taking longer than the time I can allocate to this paper. I have spent more time discussing and reviewing this paper than with the rest of the papers combined, I simply don't have the time (nor I think this is the optimal medium) to continue this discussion. I was asked for a review, I gave it, and I continued the discussion for a while. I don't have time or energy to explain envelope theorems and gradient estimators to the authors.

---

> > > > > > > ### Author Response · Authors · 2018-11-28
> > > > > > > **We really appreciate your effort. Thank you very much.  And we enjoy the discussion with you.**
> > > > > > >
> > > > > > > We really appreciate your effort. Thank you very much and we really enjoy the discussion with you.
> > > > > > >
> > > > > > > We are very rigorous in the rebuttal and we have very carefully checked related materials. In particular, we read the Wikipedia and the proof of theorem 3 line by line, as well as the paper you mentioned: ‘‘Sobolev training of neural networks’’.
> > > > > > >
> > > > > > > We wish the reviewer could consider our argument in depth and understand what we said, instead of denying it without reliable arguments. We are glad to continue the discussion at any time.
> > > > > > >
> > > > > > > If the reviewer really does not want to continue the discussion, we will respond to all concerns that the reviewer posted and leave the judgment to other reviewers, chairs, and the public.
> > > > > > >
> > > > > > > We wish the reviewer could spare a little bit more time to respond to our final three questions.
> > > > > > > 1. Do you still think there is any technical correctness issue in our paper? If so, please specify and list them.
> > > > > > > 2. Would you please provide a comment to each of our listed contributions (4+4)?
> > > > > > > 3. Would you please respond to one key remaining issue in our last response which we paste as follows?
> > > > > > >
> > > > > > > Q: It's widely known that the gradient of an estimator is not necessarily an estimator of the gradient unless one imposes constraints such as bounded Sobolev norms. The same problem appears in e.g. actor-critic formulations or other problems like synthetic gradients (see Sobolev training of neural networks for instance).
> > > > > > >
> > > > > > > >> We have checked that paper ‘‘Sobolev training of neural networks’’. Could you specify which sentence or theorem in that paper provides any support for your claim? It is simply making use of the trivial fact that we highlighted in ‘‘2. What is trivial and what is not?’’ to improve the training. It definitely makes sense and has nothing to do with our arguments here.
> > > > > > >
> > > > > > > Thank you very much. We really enjoy and appreciate the discussion with you.
> > > > > > >
> > > > > > > -------------------------------------------------
> > > > > > >
> > > > > > > Specific response to your reply, in case you would continue the discussion:
> > > > > > >
> > > > > > > Q1: NO, one can't if the dual space itself X varies on theta.
> > > > > > >
> > > > > > > >> Why? Could you provide your argument or reference?
> > > > > > >
> > > > > > > Q2: Derivative of the loss wrt the critic is 0 when the critic is optimal so the corresponding term to the changes of the critic vanishes.
> > > > > > >
> > > > > > > >> This is the key mistake in the proof. Actually, we have already pointed this out in the last response. But it seems more details are necessary. Please check the following.
> > > > > > >
> > > > > > > >> In Envelope theorem, it is critical to writing the objective as a simple function of the parameter, where all other variables need to be written as functions of the parameter. In particular, the critic function in Theorem 3 of Wasserstein GAN is not written as a function of the parameter, which leads to its false conclusion.
> > > > > > >
> > > > > > > >> More specifically, "the derivative of the loss wrt the critic is 0" does not mean "the derivative of the critic wrt the parameter" is zero. (Section 2.3 provides an instance; please recheck it.)
> > > > > > >
> > > > > > > >> If it is still not clear to you, the arguments in "2. What is trivial and what is not? " might also help. Please have a check.
> > > > > > >
> > > > > > > -------------------------------------------------
> > > > > > >
> > > > > > > Clarification (in case there exists any possible confusion):
> > > > > > >
> > > > > > > The ''envelope theorems and gradient estimators'' are related materials that the reviewer points out, with which the reviewer claims one of our contributions is meaningless. And we are correcting the reviewer such that him/she can correctly understand our contribution about whether ''criticizing a well-defined distance metric does not necessarily guarantee the convergence of GANs'' is meaningful or not.

---

> > > > > > > > ### Author Response · Authors · 2018-12-03
> > > > > > > > **More detailed arguments on why "Theorem 3 in Wasserstein GAN is a false Theorem" are provided.**
> > > > > > > >
> > > > > > > > It seems the reviewer still does not get our arguments on why "Theorem 3 in Wasserstein GAN is a false Theorem". We have polished our arguments and provided more details. Please check them out in our response to Q2, in the above reply.
> > > > > > > >
> > > > > > > > Though we don't think "Theorem 3 in Wasserstein GAN" and "envelope theorem" are very relevant to our paper, we'd like to fully address any concern from the reviewer.

---

> > > ### Author Response · Authors · 2018-11-27
> > > **We truly appreciate your feedback. Let’s continue the discussion. (1/2)**
> > >
> > >
> > > Q: "We would also like to provide an intuitive explanation on why “f(x) - f(y) <= d(x, y) for all x ~ Pr, y ~ Pg” is enough. " This is obvious.
> > >
> > > >> According to your response, we think we have reached an agreement that the new dual form of Wasserstein distance we provide in the paper is correct. In particular, Theorem 6 is correct, which states that ‘‘our new dual form, the dual form with Lipschitz condition and Wasserstein distance in primal form are equivalent’’. This is the foundation. Let’s make it very sure.
> > >
> > > Q: This is not a flaw of the Wasserstein distance, but the gradient with respect to your particular dual formulation is not. If the authors want to make the claim that not all dual formulations satisfy an envelope theorem, this is the claim the authors should be making, not that the Wasserstein distance itself is broken.
> > >
> > > >> In the current version, whenever we say Wasserstein distance does not guarantee the convergence, it is always associated with terms like ‘‘in our new dual form’’ or “in the dual form with compacted constraint”. We think it will bring no confusion. We would appreciate if you could have a check on it.
> > >
> > > >> Given that Wasserstein distance in some dual form does not guarantee the convergence, it is valid to reach our core claim that a well-defined distance metric does not necessarily guarantee the convergence of GANs. Systematic arguments are included in Section 2.4.
> > >
> > > >> We really appreciate the final solution of Wasserstein GAN, and we think it is an amazing work that brings GANs into the new stage. Therefore, we think it is important to correct the key argument in the Wasserstein GAN paper and make people realize that a well-defined distance metric does not necessarily guarantee the convergence of GANs. Hopefully, people would **stop working on new metric for GANs that does not provide reliable gradient and start to work on new mechanisms that provide reliable gradient for GAN training**.
> > >
> > > Q: The gradient of the Wasserstein distance wrt the generator's parameters itself is well defined and well behaved.
> > >
> > > >> More strictly, it should be stated as ‘‘the gradient of Wasserstein distance in the dual form that has Lipschitz-continuity condition ...’’. Given that we now know that the dual form of Wasserstein distance is not unique, we need to be more precise.
> > >
> > > Q: This is why figure 1, b) doesn't make any sense, since the Kantorovich dual is defined in the *entire* space.
> > >
> > > >> In Figure 1b, we are talking about our new dual form of Wasserstein distance, not the ‘‘Kantorovich dual’’. We have specified in the caption that it is about the new dual form of Wasserstein distance. If necessary, we can further add the specification in the sub-caption.

---

> ### Author Response · Authors · 2018-11-24
> **The authors are looking forward to your feedback.**
>
> Dear reviewer,
>
> We have provided the proof and hope that the proof and associated arguments can address your concerns about ‘‘several false statements in this paper’’.
>
> If possible, we would like to have more discussions with you. The discussions/suggestions would be helpful to us. And we are pleased to improve the paper according to your feedback if the arguments are not clear enough.
>
> Thanks for reading this message. We will be grateful for any feedback you provide.
>
> Respectfully,
> The authors

---

### Official Review · AnonReviewer1 · 2018-11-04
**Lipschitzness of the discriminator is more critical than the choice of the divergence**

**Rating:** 6
**Confidence:** 4

**Review:**

The authors study the fundamental problems with GAN training. By performing a gradient analysis of the value surface of the optimal discriminator, the authors identify several key issues.

In particular, for a fixed GAN objective they consider the optimal discriminator f* and analyze the gradients of f* at points x ~ P_g and x~P_d. The gradient decouples into the magnitude and direction terms. In previous work, the gradient vanishing issue was identified and the authors show that it is fundamentally only controlling the magnitude. Furthermore, controlling the magnitude doesn’t suffice as the gradient direction itself might be non-informative to move P_g to P_d. The authors proceed to analyze two cases: (1) No overlap between P_g and P_d where they show that the original GAN formulation, as well as the Wasserstein GAN will suffer from this issue, unless Lipschitzness is enforced. (2) For the case where P_g and P_d have overlap, the gradients will be locally useful which the authors identify as the fundamental source of mode collapse.

The main theoretical result suggests that (1) penalizing the discriminator proportionally to the square of the Lipschitz constant is the key -- the choice of divergence is not. This readily implies that pure Wasserstein divergence may fail to provide useful gradients, as well as that other divergences combined with Lipschitz penalties (precise technical details in the paper) might succeed. Furthermore, it also implies that one can mix and match the components of the objective function for the discriminator, as long as the penalty is present, giving rise to many objectives which are not necessarily proper divergences. Finally, one can explain the recent success of many methods in practice: While the degenerate examples showing deficiencies of current methods can be derived, in practice we implement discriminators as some deep neural networks which induce relatively smooth value surfaces which in turn make the gradients more meaningful.

Pro:
- Clear setup and analysis of the considered cases. Interesting discussion from the perspective of the optimal discriminator and divergence minimization. The experiments on the toy data are definitely interesting and confirm some of the theoretical results.
- A convincing discussion of why Wasserstein distance is not the key, but rather it is the Lipschitz constant. This brings some light on why the gradient penalty or spectral normalization help even for the non-saturating loss [2].
- Discussion on why 1-Lip is sufficient, but might be too strong. The authors suggest that instead of requiring 1-Lip on the entire space, it suffices to require Lipschitz continuity in the blending region of the marginal distributions.

Con:
- Practical considerations: I appreciate the theoretical implications of this work. However, how can we exploit this knowledge in practice? As stated by the authors, many of these issues are sidestepped by our current inductive biases in neural architectures.
-  Can you provide more detail on your main theorem, in particular property (d). Doesn't it imply that the discriminator is constant?
- Which currently known objectives do not satisfy the assumptions of the theorem?
- The work would benefit from a polishing pass.

========
Thank you for the response. Given that there is no consensus on the questions posed by AnonReviewer2, there will be no update to the score.

---

> ### Author Response · Authors · 2018-11-16
> **Response to Reviewer 1**
>
> Thanks for your constructive feedback.
>
> Q: How can we exploit this knowledge in practice? As stated by the authors, many of these issues are sidestepped by our current inductive biases in neural architectures.
>
> >> Although fine-tuning the neural architectures could possibly make GANs work, it is also the fundamental reason why GANs are hard to train and easily broken. Investigating theoretically-sound GAN formulation is thus important from our perspective.
>
> >> Our work actually has a few practical implications. For example, we tested a few new objectives which are also sound according to our theorems. And we found that, compared with W-distance, the outputs of the discriminator with some new objectives are more stable and the final qualities of generated samples are also consistently higher than those produced by W-distance.
>
> Q: Can you provide more detail on your main theorem, in particular, property (d). Doesn't it imply that the discriminator is constant?
>
> >> The main theorem describes the properties of the optimal discriminative function under Lipschitz constraint. In a nutshell, the discriminator would stretch the value surface by adjusting the value of f(x) for each x (from P_g and P_r) till |f(x)-f(y)|=|x-y| for some y. The key insight behind the properties is that at optimum, for every x, f*(x) must hold a zero gradient with respect to the objective or it is bounded by the Lipschitz constraint.
>
> >> Property-(d) states that at the only Nash Equilibrium, it holds P_g = P_r. And it can also be proved that at the Nash Equilibrium state, k(f*) must be zero. k(f*)=0 means the discriminative function is constant, but it only appears at the converged state P_g = P_r. Constant discriminative function at convergence is actually a good/desired property, where the generator receives zero gradients and hence will not shift from the convergence point.
>
> Q: Which currently known objectives do not satisfy the assumptions of the theorem?
>
> >> There exists a few practically used instances of GAN objectives that do not satisfy the assumptions of the theorem. For example, the objective of Least-Square GAN and the hinge loss used in [1][2][3]. More generally, any objective that ‘‘holds a zero gradient at certain point’’ does not satisfy the assumptions of the theorem (check Eq. 12).
>
> [1] Geometric GAN
> [2] Energy-based Generative Adversarial Network
> [3] Spectral Normalization for Generative Adversarial Networks

---

> ### Author Response · Authors · 2018-12-07
> **Thank you very much for your update.**
>
> We sincerely thank you for considering our discussion with AnonReviewer2. With all due respect, from the authors’ perspective, AnonReviewer2 has made enormous mistakes; and we are not able to get into a consensus with AnonReviewer2, because he/she refuses further discussion. The authors thus hope there is an official judgment. A formal request is posted in the topmost comment.

---

### Author Response · Authors · 2018-11-16
**We have provided the proof on the new dual form of Wasserstein distance.**

Dear reviewers,

We have provided detailed proofs of our new dual form of Wasserstein distance (in Appendix I). We hope the proofs and the associated detailed explanations can address the concern about “the possible wrong in our key arguments”.

According to the reviewers’ feedback, we have extensively revised the paper. The arguments are much more clear now. In particular, we have revised all statements about the failure of Wasserstein distance to make them strictly refer to the new dual form. We have also made the relationship among Lipschitz condition, Wasserstein distance, and the new dual form more clear. We believe there is no confusion in our statements now.

The key points are summarized as follows: (i) the dual form of Wasserstein distance can be written in a more compact manner, where the constraint is looser than Lipschitz condition. (ii) if using Wasserstein distance with the new dual form, it suffers from the convergence issue, where \nabla_x f*(x) is ill-behaving. (iii) the above observations indicate that a well-defined distance metric does not necessarily guarantee the convergence of GANs. (iv) we prove that Lipschitz condition is a general key to solve the non-convergence problem of GANs, which works with a family of GAN objectives (detailed in Eq. 12) and is not limited to Wasserstein distance.

Detailed comments from each reviewer have been addressed individually. Thanks a lot for these constructive feedbacks. And special thanks to AnonReviewer3 who have meticulously checked our notations and formulations and provided detailed feedback, which helps us a lot in improving this manuscript.

If the reviewers have further concerns, we would appreciate further discussions.

---

### Author Response · Authors · 2018-11-28
**Contribution clarification.**


We sincerely thank the reviewers for their constructive comments and valuable suggestions. Since the contributions of this paper are one of the topics that we have discussed with the reviewers. We summarize the contributions of this submission as follows for ease of reference:

Major Contributions:

1. In this paper, we have shown that in many GANs, the gradient from the optimal discriminative function is not reliable, which turns out to be the fundamental cause of failure in training of GANs. Instances that are explicitly discussed in the paper include Original GAN, Least-Square GAN, Fisher GAN, and GAN with a new dual from of Wasserstein distance.*

2. We have highlighted in this paper that a well-defined distance metric does not necessarily guarantee the convergence of GANs. Because in typical GAN settings, the optimal discriminative function is only a helper of the estimator that estimates some distance metric, whose gradient does not necessarily reflect the gradient of the distance metric (including JS divergence, Fisher IPM, Wasserstein distance in a new dual form). Hence, we call on researchers to pay more attention to the design of the optimal discriminative function (and its gradient) when introducing new GAN objectives.

3. We have proved in this paper that the Lipschitz-continuity condition as a general solution to make the gradient of the optimal discriminative function reliable, and characterized the necessary condition where Lipschitz condition ensures the convergence, which leads to a broad family of valid GAN objectives under Lipschitz condition, where Wasserstein distance is one special case.

4. We have tested several new objectives with experiments, which are also sound according to our theorems. And we found that, compared with Wasserstein distance, the outputs of the discriminator with some new objectives are more stable and the final qualities of generated samples are also consistently higher than those produced by Wasserstein distance.

* It is not limited to these GANs. A straightforward extension as mentioned in the paper is that all GANs where the optimal discriminative function f^*(x) at x is only related to P_g(x) and P_r(x), e.g., f-GAN.

---

> ### Author Response · Authors · 2018-11-28
> **Contribution clarification.**
>
>
> Minor Contributions:
>
> 1. We have shown that the locality of the optimal discriminative function and its gradient in traditional GANs is an intrinsic cause to mode collapse.
>
> 2. We have conducted a fairly systematic study on how the hyper-parameters influence the optimal discriminative function in traditional GANs (Appendix A), which explains why traditional GANs are sensitive to hyper-parameters, hard to train and easily broken.
>
> 3. We have provided a new technique for implementing the Lipschitz-continuity condition in Appendix E, which, according to our experiments, can achieve the theoretically optimal discriminative function in many cases where the gradient penalty or spectral normalization can not.
>
> 4. We have provided a new dual form of Wasserstein distance (proved in Appendix I).

---

### Author Response · Authors · 2018-12-07
**We hope we can get an official judgment on our discussion with AnonReviewer2.**

We just notice that AnonReviewer1 provided a feedback on our discussion with AnonReviewer2. It seems the reviewers did not get into a consensus on our discussion with AnonReviewer2.

1) However, from the authors’ perspective, AnonReviewer2 made enormous mistakes and the authors have convinced the AnonReviewer2 that there is no technical correctness issue in this paper. Because at the final stage of the discussion, AnonReviewer2 only tried to claim "showing the failure of a new dual form of Wasserstein distance is meaningless", i.e., something about contribution.

2) Furthermore, from the authors’ perspective, most of the arguments provided by AnonReviewer2, with which to claim "showing the failure of a new dual form of Wasserstein distance is meaningless", are also false. The authors have provided corresponding arguments showing that AnonReviewer2 has many misunderstandings on related materials and also clarified that the contribution of this paper is much more than just "showing Wasserstein distance in new dual form might also fail". AnonReviewer2 refuses further discussion.

As one of the top confluences like ICLR, we think it should be able to judge: are the authors making mistakes, or is the AnonReviewer2 making mistakes? We hope we can get an official judgment on whether there is technical correctness issue in the paper and whether the contribution is significant.

Respectfully,
The authors

---

### Meta-Review · Area_Chair1 · 2018-12-14
**Intersting theoretical analyis of a compact dual form of the Wasserstein distance which however is not widely used in literature.**

**Confidence:** 4
**Recommendation:** Reject

**Metareview:**

The paper investigates problems that can arise for a certain version of the dual form of the Wasserstein distance, which is proved in Appendix I. While the theoretical analysis seems correct, the significance of the distribution is limited by the fact, that the specific dual form analysed is not commonly used in other works. Furthermore, the assumption that the optimal function is differentiable is often not fulfilled neither. The paper would herefore be significantly strengthen  by making more clear to which methods used in practice the insights carry over.

---

> ### Author Response · Authors · 2018-12-21
> **Thanks.**
>
> We sincerely thanks all ICLR reviewers and Area Chair for their effort in reviewing this paper and all constructive comments. But we should highlight that this paper is not a "theoretical analysis of a compact dual form of the Wasserstein distance".
>
> The core of this paper is, however, that we show that "a well-defined distance metric does not necessarily guarantee the convergence of GANs" and this is the fundamental cause of failure in training of GANs. And we show that Lipschitz-continuity condition is a general solution to this problem, and characterized the necessary condition where Lipschitz condition ensures the convergence. (see details in the paper.)